# Health Economic Aspects of Childhood Excess Weight: A Structured Review

**DOI:** 10.3390/children9040461

**Published:** 2022-03-24

**Authors:** Olu Onyimadu, Mara Violato, Nerys M. Astbury, Susan A. Jebb, Stavros Petrou

**Affiliations:** 1Nuffield Department of Primary Care Health Sciences, University of Oxford, Radcliffe Observatory Quarter, Woodstock Road, Oxford OX2 6GG, UK; olu.onyimadu@phc.ox.ac.uk (O.O.); nerys.astbury@phc.ox.ac.uk (N.M.A.); susan.jebb@phc.ox.ac.uk (S.A.J.); 2Nuffield Department of Population Health, University of Oxford, Old Road Campus, Oxford OX3 7LF, UK; mara.violato@dph.ox.ac.uk

**Keywords:** childhood, obesity, cost-of-illness, cost-effectiveness, economic evaluation, utilities, socioeconomic, food price, human capital, structured review

## Abstract

An economic perspective is crucial to understand the broad consequences of childhood excess weight (CEW). These can manifest in the form of elevated health care and societal costs, impaired health status, or inefficiencies in the allocation of resources targeted at its prevention, management, or treatment. Although existing systematic reviews provide summaries of distinct economic research strands covering CEW, they have a restricted focus that overlooks relevant evidence. The overarching aim of this structured review was to update and enhance recent key reviews of four strands of economic evidence in this area, namely, (1) economic costs associated with CEW, (2) health utilities associated with CEW, (3) economic evaluations of interventions targeting CEW, and (4) economic determinants and broader consequences of CEW. Our de novo searches identified six additional studies for the first research strand, five studies for the second, thirty-one for the third, and two for the fourth. Most studies were conducted in a small number of high-income countries. Our review highlights knowledge gaps across all the research strands. Evidence from this structured review can act as data input into future economic evaluations in this area and highlights areas where future economic research should be targeted.

## 1. Introduction

The global prevalence of childhood obesity has increased markedly, although regional and between-country variations remain considerable [1,2]. A ranking of countries by sociodemographic index (SDI) levels indicates that the prevalence of childhood obesity is greater in high-income countries [3]. Nonetheless, low SDI countries saw a 20% increase in the prevalence of childhood obesity between 1980 and 2015 [3]. Within countries and geographical regions, significant variations in prevalence have also been reported among gender, age, ethnicity, and socioeconomic status (SES) subgroups [4,5,6,7].

Childhood overweight and obesity (defined by UK-WHO growth charts as a body mass index (BMI) above the 91st and 98th centiles, respectively) [8], collectively referred to as childhood excess weight (CEW), are major public health concerns characterised by multi-faceted health consequences. In the short term, children with excess weight are more likely to suffer mental illnesses than their healthy-weight counterparts and tend to report poorer outcomes in education, with potentially greater health care costs during childhood and knock-on effects on human capital development and societal costs [2,9]. Long-term health consequences of CEW, such as increased risks of cardiometabolic diseases in adulthood, impose substantial health care and indirect costs on society [10]. Obesity-related externalities that manifest as costs borne by society are often the basis for government-based intervention [11].

The discipline of economics offers a range of methods that help understand values and behaviours around the utilisation of health care services related to CEW, and the efficiency of interventions and programmes targeted at the prevention, management, and treatment of CEW [10,12]. Four recently published systematic reviews [9,13,14,15] summarise the literature covering distinct economic perspectives on CEW, namely, economic costs associated with CEW, health state utility values (HSUVs) associated with CEW, economic evaluations targeting CEW, and broader economic consequences of CEW.

A systematic review by Hamilton et al. 2018 [13] quantified the economic costs of CEW (research strand 1). In their meta-analysis, Hamilton and colleagues estimated the mean total lifetime cost (health care costs and the value of productivity losses) for boys and girls with obesity to be EUR 149,206 and EUR 148,196, respectively (2014 euros; GBP 142,258 and GBP 141,295, respectively, in 2020 pounds sterling [16]). However, the study was constrained by only including studies that estimated costs over a lifetime horizon. Furthermore, in some included studies [17,18,19], the baseline age for weight measurement was 20 years. This did not appear to meet the inclusion criterion stated by Hamilton and colleagues [13] about study baseline weight needing to be measured in childhood—an indication that the estimated economic costs may not be attributable to weight gain during childhood. In addition, the synthesised productivity impacts were driven by only two studies [20,21], and may have overestimated their derived indirect costs.

Brown et al. 2018 [14] conducted a systematic review of studies that synthesised HSUVs or health utilities (values that capture individual perceptions of health states using preference-based methods [22,23]), which can act as inputs into cost-utility analyses (CUAs) of interventions targeting CEW (research strand 2). The authors meta-analysed HSUVs, estimating mean utility values of 0.85, 0.83, 0.82, and 0.83 for healthy weight, overweight, obese, and overweight/obese states, respectively. None of the studies included in the review by Brown et al. [14] were based on longitudinal study designs, which are crucial for exploring potential reverse causation.

Zanganeh et al. 2019 [15] undertook a systematic review of methods, study quality, and the results of economic evaluations of interventions targeting CEW (research strand 3). The most common intervention category was behavioural, and the study approaches were predominantly preventive. The included studies were conducted predominantly in high-income countries. Most of the interventions evaluated were cost-effective, but evidence synthesis was deemed challenging due to methodological heterogeneity. This review excluded pharmacological and surgical interventions.

Finally, Segal et al. 2021 [9] conducted a systematic review that assessed the impact of CEW on human capital development outcomes (research strand 4). Among the included studies, cognitive performance, captured through test scores, was the most researched outcome. In comparison, educational attainment (measured through grade progression and college completion) and labour market outcomes (measured through wages later in life) were under-researched. Evidence of lower cognitive function and educational attainment due to CEW seemed persuasive, particularly in girls. The authors omitted two articles [20,21] comprising a total of five studies that investigated labour market and educational attainment outcomes in childhood populations.

The purpose of this study was to provide a structured review of economic aspects of childhood excess weight, updating the previous reviews and extending to cover (1) economic costs associated with CEW, (2) HSUVs associated with CEW, (3) economic evaluations of interventions targeting CEW, and (4) economic determinants and broader consequences of CEW. Ultimately, we aim to draw concise summative conclusions across the identified strands of evidence and help prioritise salient economic research questions in the field of CEW.

## 2. Materials and Methods

We identified the above-mentioned systematic reviews and related research strands after conducting preliminary searches in PubMed and Google Scholar on key health economic concerns surrounding CEW. In this section, we document our methods for updating and augmenting the reviews and summarising evidence within each research strand. Throughout this study, the terms ‘children’ and ‘childhood’ refer to infants, adolescents, young people, and all persons between the ages of 0 and 18 years.

### 2.1. Economic Costs Associated with Childhood Excess Weight

The previously cited review by Hamilton et al. [13] searched eight databases for cost-of-illness studies published from 1 January 2000 to 20 February 2016, but only the search strategy for PubMed was reported. We adapted the PubMed search strategy used by the authors and additionally searched Medline via Ovid, covering studies published from 2016 to 14 January 2021 (Appendix A). Our adaptation of the eligibility criteria included studies where the age upon study entry fell during childhood; that compared children of different weight categories, e.g., healthy weight children versus children with obesity; and that reported economic costs across any time horizon. We excluded studies if the age at study entry exceeded 18 years; outcomes were not expressed in terms of economic costs, expenditures, or charges; or were not published in English.

We identified and extracted key assessment items from the individual studies identified by our de novo searches as well as studies identified by the original review by Hamilton and colleagues [13] that met our refined inclusion criteria. To facilitate comparisons between studies, we extracted cost outcomes from individual studies and updated them using a web-based tool for adjusting cost estimates (the CCEMG—EPPI-Centre Cost Converter), expressing cost values in 2020 prices and in British pounds sterling (GBP). The CCEMG—EPPI-Centre Cost Converter uses national price indices to inflate costs to a common price date and purchasing power parities (PPPs) provided by the International Monetary Fund (IMF) for currency conversions [16]. We assessed the quality of the included economic cost studies using 19 relevant items in the Consolidated Health Economic Evaluation Reporting Standards (CHEERS) statement [24]. To ensure that the reviewed studies were scored fairly, we estimated separate denominators for the individual studies based on the relevance of individual CHEERS checklist items to each study design.

### 2.2. HSUVs Associated with Childhood Excess Weight

We adopted the search strategies developed in the systematic review by Brown et al. [14]. We then searched Medline via Ovid for studies published from 2017 to 19 January 2021. The contents of these search strategies and the search outputs can be found in Appendix A and Appendix A, respectively.

We used the same eligibility criteria applied in the original review by Brown et al. [14], namely, studies published in peer-reviewed journals reporting primary data collection of utility values, with results reported by weight status; studies conducted in childhood populations; and studies reported in the English language.

We identified and extracted key assessment items from the individual studies in our de novo search as well as studies in the original review by Brown and colleagues [14] that met the inclusion criteria. We assessed the quality of the studies using the key items recommended by the checklist developed by Papaioannou et al. 2013 [25] for assessing studies reporting HSUVs. These included reporting of response rates to instrument used, loss to follow-up, missing data, and the potential effects of these items on the validity of the estimates; sample size; respondent selection and recruitment; study inclusion and exclusion criteria; and appropriateness of measure for the population or patient group.

### 2.3. Economic Evaluations of Interventions Targeting Childhood Excess Weight

The previous review by Zanganeh and colleagues [15] had searched for economic evaluations of interventions targeting CEW published between January 2001 and April 2017. We adapted their search strategy and searched Medline via Ovid for studies published from 2017 to 17 January 2021 (Appendix A). Our review encompassed economic evaluations of interventions targeting CEW regardless of the type of economic evaluation, or the approach underpinning the economic evaluation (e.g., randomised controlled trial (RCT), observational study with individual-level data, decision-analytic model, quasi-experimental design, etc.). Our eligibility criteria included studies if they targeted children at the time of intervention; evaluated treatment, prevention, or management interventions targeting CEW, including pharmacological or surgical; contained at least one comparator, such as usual care or ‘no treatment’ or ‘no intervention’; and reported economic evaluation outcomes, such as incremental cost-effectiveness ratios (ICERs) expressed in metrics such as incremental cost per quality-adjusted life year (QALY) gained, incremental cost per unit change in health effect (e.g., incremental cost per change in BMI z-score), or net monetary benefits. Studies were excluded if they were not focused on tackling excess weight (overweight or obesity) in childhood; not an economic evaluation; the title and abstract were not published in the English language; a protocol study; or a review article not containing primary research evidence.

We identified and extracted key assessment items from the individual studies in our de novo search as well as studies in the original review by Zanganeh and colleagues [15] that met our inclusion criteria. The quality of each study was assessed using the CHEERS statement [24]. We used only applicable CHEERS items in the assessment of individual studies. For instance, decision-analytic modelling items (item numbers 15 and 16) were not relevant for within-trial economic evaluations. To ensure that the reviewed economic evaluations were scored fairly, we estimated separate denominators for the individual studies based on the relevance of individual CHEERS checklist items to each study design.

#### Definitions and Taxonomies

Interventions were categorised as behavioural, environmental, pharmacological, surgical, or policy-focussed, or a combination of any of these individual categories. Behavioural interventions refer to procedures and practices aimed at changing the individual’s behaviour. Environmental interventions aim to modify or adapt the environment. Pharmacological interventions include prescribed or over-the-counter pharmaceutical drugs or medicines. Surgical interventions are medical and dental procedures that involve surgery. Interventions classified as policy interventions can be applied, proposed, or modelled at the population level through legislation or fiscal actions and may affect advertising, taxes, levies, and subsidies.

The study approaches were categorised as prevention, treatment, or management-focussed, or a combination of any of these individual categories. A preventive approach was an intervention targeted at normal-weight children or children with overweight to prevent progression to obesity. An approach was defined as management-based if the intervention aimed at keeping the excess weight condition in control but did not necessarily treat it. An approach that targeted a mixed population of normal weight and excess weight children could have been described as both preventive and treatment-based or preventive and management-based.

An intervention was described as cost-effective if the authors of the study carried out cost-effectiveness or cost-utility analysis and deemed the intervention cost-effective with reference to the cost-effectiveness threshold in the country in which the study was conducted. An intervention was described as potentially cost-effective if the authors deemed the intervention cost-effective but did not make explicit reference to a cost-effectiveness threshold or performed a different type of economic evaluation (e.g., cost-benefit or cost-consequences analysis). An intervention was also described as potentially cost-effective if the authors expressed concerns about uncertainties, such as implementation, despite considering it cost-effective. Evidence was described as inconclusive if the authors were unable to conclude that it was cost-effective or if there was methodological reason to disagree with the judgment of the authors. An intervention was described as not cost-effective if the authors of the study carried out cost-effectiveness or cost-utility analysis and deemed the intervention not cost-effective with reference to the cost-effectiveness threshold in the country in which the study was conducted, or if the costs exceeded the benefits in cost-benefit analyses. Generally, all statements about the cost-effectiveness of interventions in this review should be treated with caution, as we did not seek access to the contributing study data and models with the view of replicating or verifying study estimates.

### 2.4. Economic Determinants and Broader Consequences of Childhood Excess Weight

We searched for published systematic reviews covering economic determinants of CEW via advanced Google Scholar, applying combinations of the search command terms ‘socioeconomic inequalities’, ‘socioeconomic status (SES)’, ‘food pricing’, ‘diet’, ‘sugar-sweetened beverages (SSBs)’, ‘soft drinks’, ‘paediatric’, ‘childhood’, and ‘youth’. We restricted our search to studies published from 2010 to 2021. We also searched the weekly updates from Obesity Intelligence (https://khub.net/web/phe-obesity-intelligence last accessed on 20 January 2021) from November 2020 to January 2021, as well as the reference lists of relevant studies.

We conducted similar advanced Google Scholar searches for systematic reviews covering the broader consequences of CEW, including the search terms ‘human capital’, ‘wages’, and ‘education’. We extracted key assessment items from the recently conducted systematic review by Segal et al. [9], as well as additional related studies identified by our searches.

## 3. Results

### 3.1. Economic Costs Associated with Childhood Excess Weight

Our searches identified 1042 papers. After screening titles and abstracts and removing duplicates, 12 potentially relevant papers were identified. Six of them were excluded because they did not meet the eligibility criteria. We applied the same eligibility criteria to the 13 studies included in the systematic review by Hamilton et al. [13] and excluded seven of them [17,18,19,20,21,26,27] because they did not meet our eligibility criteria on baseline exposure age, estimated outcomes, or language in the full text publication. We included a total of 12 studies in our structured review, comprising six studies from the review by Hamilton et al. [28,29,30,31,32,33] and six studies from our de novo searches [34,35,36,37,38,39].

Table 1 aggregates studies based on shared characteristics. All the studies identified in this review were conducted in high-income countries: seven in the United States (U.S.) [28,29,30,31,37,38,39], two in Germany [32,33], two in Australia [34,36], and one in The Netherlands [35]. Six of the economic cost studies [28,29,30,32,33,39] were based on decision-analytic models such as microsimulation models or state transition models, whilst five studies [31,34,36,37,38] were based on regression modelling of longitudinal datasets or panel data analysis. Only three studies [29,33,37] estimated indirect costs, and the study time horizon ranged from one year to lifetime across all studies. In Table 2, we report 12 key assessment items for individual studies, including type of study design, age range upon study entry, study time horizon, compared weight status groups, type of economic costs estimated, discount rates, estimated economic costs, and the overall quality score (%). The quality of studies ranged from 81% to 100% for the included studies, based on relevant items of the CHEERS checklist [24].

#### 3.1.1. Summary of Study Results—Excess Direct Costs

The studies estimating direct costs (resources consumed in the treatment or management of children with excess weight) associated with CEW differed in terms of the age at which weight status was measured and the study time horizon for cost estimation. The age ranges at study entry varied between 2 and 18 years. Hayes et al. [34] and Lightwood et al. [29] estimated direct costs between 2 years and 5 years, and 2 years and 19 years, respectively. The baseline age for children in the study by Schell et al. [39] was 18 years. A discount rate of 3% was applied in most of the 11 studies that estimated direct costs where the study or model time horizon exceeded 1 year. Two studies [34,38] with time horizons of 3 years and 10 years of follow-up did not report any discount rates.

Statistical methods for cost estimation broadly followed either an econometric/regression analysis approach or a decision-analytic modelling approach. However, these approaches are umbrella terms that describe a collection of methodological approaches. The use of regression-based analyses to estimate costs was predominantly applied in studies with a time horizon of three years or less [31,34,36,37,38]. Among these studies, the two-part model specification, recommended as the choice statistical model for handling the substantial skewness and mass of zeros often seen in health care cost data [40,41], was mostly applied, or, when not applied, duly justified [34]. Two recent studies used an instrumental variable (IV) approach to assess the economic burden in Australia and the U.S. [36,37]. Both studies utilised the BMI of biological parents as instruments for child BMI. Decision-analytic studies were conducted mostly through state-transition modelling [42] and tended to estimate costs over a lifetime horizon [28,30,32,39], except for the study by Lightwood et al. [29], where the time horizon adopted was 30 years.

**Table 2 children-09-00461-t002:** Descriptive analysis of the key assessment items for individual economic cost studies.

First Author and Year of Publication	Country	Type of Study Design (or Decision-Analytic Model if Applicable)	Age Range upon Study Entry	Study Time Horizon	Exposures/Measures of Weight Status Compared	Type of Economic Cost(s) Estimated	Currency Unit (Price Year)	Discount Rates	Sensitivity Analyses (Further Analytical Approaches)	Estimated Economic Costs *	Quality Score (%)
Fernandes, M.M. 2009 [28]	United States	Cohort (Monte Carlo) simulation model	9 years	Lifetime	Obese versus normal-weight	Direct costs	U.S. dollars (2008)	3% annually	DSA and PSA	Excess lifetime costs of USD 12,047 and USD 15,639 per boy and girl, respectively	18.5/19 (97%)
Lightwood, J. 2009 [29]	United States	Markov model	12 to 19 years	30 years	Obese versus normal-weight	Direct and indirect costs	U.S. dollars (2007)	3% annually	DSA	Projected (2020–2050) cumulative excess direct, indirect, and total costs: USD 46 billion, USD 208 billion, and USD 254 billion, respectively	17.5/19 (92%)
Trasande, L. 2010 [30]	United States	Cohort simulation model	12 years	Lifetime	Obese and overweight versus normal-weight	Direct costs	U.S. dollars (2005)	3% annually	DSA	Total direct medical expenditures (child and adult) attributable to childhood overweight/obesity for male and female is USD 2.94 billion and USD 3.3 billion, respectively	16.5/19 (87%)
Ma, S. and Frick, K. D. 2011 [31]	United States	Panel data analysis (two-part regression model)	6 to 17 years	1 year	Obese versus normal-weight	Direct costs	U.S. dollars (2006)	3% annually	DSA (Controlled for covariates)	Excess annual medical expenditure: USD 264 per capita	15/17 (88%)
Sonntag, D. 2015 [32]	Germany	Markov model	3 to 17 years	Lifetime	Obese and overweight versus normal-weight	Direct costs	Euros (2010)	3% annually	DSA	Excess lifetime direct costs (excess weight): EUR 7028 and EUR 4262 per girl and boy, respectively	19/19 (100%)
Sonntag, D. 2016 [33]	Germany	Markov model	3 to 17 years	Lifetime	Obese and overweight versus normal-weight	Indirect costs	Euros (2010)	3% annually	DSA and PSA	Excess lifetime indirect costs (excess weight): EUR 2445 and EUR 4209 per girl and boy, respectively	19/19 (100%)
Hayes A. 2016 [34]	Australia	Longitudinal cohort analysis (multivariable regression analyses)	2 to ≤5 years	3 years	Obese and overweight versus normal-weight	Direct costs	Australian dollars (2014)	Not stated	DSA (Controlled for covariates)	Excess mean 3-year health care costs: AUD 1608 and AUD 93 for an obese and overweight child, respectively	16/17 (94%)
Wijga, A.H. 2018 [35]	The Netherlands	Longitudinal birth cohort analysis (Wilcoxon–Mann–Whitney test for statistical significance)	14 to 15	1 year	Overweight (including obesity) and non-overweight	Direct costs	Euros (2011)	NA	None reported	Mean excess annual health care expenditure: EUR 221	13/16 (81%)
Black, N. 2018 [36]	Australia	Longitudinal panel analysis (two-part regression model with IV estimator as base case)	6 to 13 years	1 year	Obese and overweight versus normal-weight	Direct costs	Australian dollars (2015)	NA	DSA (Controlled for covariates)	Excess annual non-hospital Medicare costs per child: AUD 63 and AUD 103 for overweight and obesity, respectively and annual medical cost due to excess weight was AUD 43 million in the full childhood population	14/16 (88%)
Biener, A.I. 2020 [37]	United States	Panel data analysis (two-part regression model with IV estimator as base case)	11 to17 years	1 year	Obesity and severe obesity versus normal-weight	Direct costs	U.S. dollars (2015)	NA	DSA (Controlled for covariates)	Per child for obesity and severe obesity, respectively, excess annual medical expenditure: USD 907 and USD 1491 and excess annual out-of-pocket expenditure: USD 25.79 and 37.36. Mean annual direct cost of obesity of USD 7.71 billion in the full childhood population	15/16 (94%)
Kompaniyets, L. 2020 [38]	United States	Longitudinal study (two-part regression model)	2 to 19 years	10 years	Primary obesity diagnosis and secondary obesity diagnosis versus no obesity diagnosis	Direct costs	U.S. dollars (2016)	Not stated	(Controlled for covariates)	Excess primary obesity diagnosis charges and costs: USD 20,879 and USD 6049, respectively. Excess secondary obesity diagnosis charges and costs: USD 3453 and USD 1359, respectively	14/17 (82%)
Schell, R.C. 2020 [39]	United States	Markov model	18 years	Lifetime	Obese versus normal-weight	Direct costs	U.S. dollars (2017)	3% annually	None reported	Excess lifetime costs: USD 22,315, USD 14,813, USD 37,329, and USD 2018 for white males, black males, white females, and black females, respectively	17/19 (89%)

DSA: deterministic sensitivity analysis; IV: instrumental variable; NA: not applicable; PSA: probabilistic sensitivity analysis. * Estimated costs are expressed in the original currency and price year.

There was marked variation in the type of economic outcomes estimated with a preponderance of studies that reported expenditures or charges rather than costs among studies conducted in the U.S. Four U.S.-based studies [30,31,37,39] reported various types of expenditure, whilst one [38] reported charges and costs. With the exception of the study by Kompaniyets et al. [38] that used the National Inpatient Sample (NIS) longitudinal database, all U.S.-based studies relied on expenditures from the Medical Expenditures Panel Survey (MEPS). A study conducted in The Netherlands by Wijga et al. [35] also estimated health care expenditures.

A further source of methodological heterogeneity between the reviewed studies involved the components and time horizon of direct costs estimated. Estimated costs were primarily direct health care costs, but direct non-medical costs were sometimes estimated. For instance, Biener et al. [37] investigated the excess annual out-of-pocket expenditure by families related to children aged 11 to 17. Excess lifetime health care costs were estimated either at an individual level in three studies [28,32,39] or at a population level in one study [30]. Mean individual-level excess lifetime costs relative to healthy weight ranged from GBP 1494 (pounds sterling, 2020 prices) for an 18-year-old black female with obesity in the study by Schell et al. [39] to GBP 13,260 (pounds sterling, 2020 prices) for a 9-year-old girl with obesity in the study by Fernandes [28]. Schell et al. [39] also estimated mean lifetime costs for white males, black males, and white females, which were GBP 16,525, GBP 10,970, and GBP 27,644, respectively (pounds sterling, 2020 prices), for children in the U.S. Fernandes [28] estimated a mean excess weight cost of GBP 10,214 for boys in the U.S., while Sonntag et al. [32] estimated a mean lifetime excess weight cost of GBP 7432 and GBP 4507 per girl and boy, respectively, in Germany (pounds sterling, 2020 prices).

Trasande [30] estimated the total direct medical expenditures accrued in childhood and adulthood attributable to childhood overweight and obesity for boys and girls at GBP 2.69 billion and GBP 3.02 billion, respectively, in the U.S. population (pounds sterling, 2020 prices) [30]. These lifetime values are not comparable to the mean annual direct cost of obesity of GBP 5.88 billion in the U.S. childhood population estimated by Biener et al. [37] or the GBP 40 billion cumulative excess direct cost projected over 30 years from 2020 to 2050 by Lightwood et al. [29] (pounds sterling, 2020 prices).

Individual-level excess mean annual direct costs were assessed in four studies and ranged from GBP 89 [36] to GBP 1829 [37] for excess weight (pounds sterling, 2020 prices). Ma and Frick [31] estimated excess annual medical expenditure due to childhood obesity in the U.S. as GBP 234, while Wijga et al. [35] estimated mean excess annual health care expenditure due to CEW in The Netherlands as GBP 217 (pounds sterling, 2020 prices). Black et al. [36] assessed excess annual non-hospital Medicare costs per child in Australia of GBP 89 for excess weight, while Biener et al. [37] estimated excess annual expenditure per child in the U.S. of GBP 1829 for obesity or severe obesity (pounds sterling, 2020 prices).

#### 3.1.2. Summary of Study Results—Excess Indirect Costs

Only two of the included studies estimated indirect costs (i.e., the values of the time and resources of the children and/or their families taken up in the treatment or management of CEW, often in the form of work disability or loss of productivity). The age ranges at study entry varied between 3 and 18 years.

Sonntag et al. [33] estimated indirect costs associated with CEW, which included the economic value of sickness absence and early retirement over a lifetime horizon in a baseline German population of children aged 3 years to 17 years. Per girl and boy, these projections were GBP 2586 and GBP 4451, respectively (pounds sterling, 2020 prices). While for boys, the estimated excess indirect costs were similar in magnitude to excess direct costs, for girls, the excess direct costs were nearly three times the excess indirect costs, potentially due to lower lifetime earnings.

Lightwood et al. [29] investigated lost productivity due to morbidity or premature death associated with children and youths in the U.S. aged 12 years to 19 years over 30 years. In contrast to the GBP 39 billion excess direct costs projected over the years 2020–2050 associated with childhood obesity, the authors projected an excess indirect cost of GBP 180 billion over the same period (pounds sterling, 2020 prices).

### 3.2. Health Utility Values Associated with Childhood Excess Weight

In our updated searches (Appendix A), we identified a total of 1066 papers. After de-duplication and title and abstract screening, 155 papers remained. We then screened the full texts, after which we identified five studies based on our eligibility criteria. Table 3 presents a summary of nine key assessment items for 14 studies: nine health state valuation studies that elicited primary utility values identified by the systematic review by Brown et al. [14] and five studies identified by our updated searches.

The overall quality score of the individual studies, as measured using the key assessment criteria outlined by Papaioannou et al. 2013 [25], ranged from 67% to 100%. The number of relevant items for the reviewed studies was either six for cross-sectional studies or seven for longitudinal studies. Response rates were not explicitly reported in four studies [46,48,49,51], while two studies by Boyle et al. [43] and Keating et al. [45] reported low response rates (33% and 50%, respectively). Details on the levels of missing data and how they were dealt with were scantily reported in many studies.

Six of the reviewed studies were conducted in Australia [45,48,50,53,55,56], four in the United Kingdom (U.K.) [43,49,51,52], two in the U.S. [44,47], and two in The Netherlands [46,54]. The targeted childhood age groups varied widely across studies but mostly encompassed teenage age groups. The youngest age considered was 5 years in three studies [44,51,53]. All nine studies identified in the systematic review by Brown et al. [14] were cross-sectional, but three of the five newly identified studies from our search were longitudinal [52,54,56]. The setting of the studies was diverse, but most of them were conducted in school settings. Other settings included clinical settings and multi-setting scenarios.

Study sample sizes varied vastly, ranging from as few as 76 children [44] to 4979 children [47]. Similarly, a wide range of utility instruments was used across the studies. Six studies used the Child Health Utility 9D (CHU-9D), making it the most frequently used instrument [49,50,51,52,55,56]. Other instruments used include the EuroQol 5D Youth (EQ-5D-Y) [43,49], EQ-5D-3L [46,54], Assessment of Quality of Life 6D (AQoL-6D) [45,48], and various versions of the Health Utilities Index (HUI) [44,47,53]. Two studies that applied the HUI3 used parental proxy reports for utility assessment [44,53]. The choice of valuation method across studies was roughly evenly distributed between time trade-off and standard gamble, but the choice of preference weights tended to derive from adult populations of the country where the study was conducted. The exception was Australia, where the preference weights were generated from adolescent populations for five studies [45,48,50,55,56].

The highest estimates of utilities associated with healthy weight were 0.90 for children aged 11 to 15 years in the U.K. [43], 0.87 for children aged six to seven years in the U.K. [50], 0.89 for children aged 11 and above in Australia [48], and 0.96 for children at age five years in Australia [53]. The lowest utility estimates were 0.83 for children aged five and six years in the U.K. [51] and 0.81 for children aged five to 18 years in the U.S. [44]. The highest estimates of utilities associated with overweight and obesity at five years were estimated in Australia at 0.96 and 0.95, respectively [53]. Killedar et al. 2019 [56] disaggregated their estimates by gender for children aged 10 to 17 in Australia. They found that the association between BMI z-score and utilities was affected by age and was statistically significant in girls but weak in boys at all ages.

### 3.3. Economic Evaluations of Interventions Targeting Children with Excess Weight

Our searches identified 282 papers. After screening titles and abstracts, and excluding duplicates, 52 potentially relevant papers were identified. We then reviewed the full-text versions of the screened papers and included 31 relevant papers. Of the 35 papers reported in the review by Zanganeh et al. [15], we excluded three papers [57,58,59] because they did not meet the inclusion criteria for this review. Our review therefore includes results for 63 studies: 32 studies from the systematic review by Zanganeh et al. plus 31 studies extracted from our updated searches.

In Table 4, the studies identified by the review are described based on shared characteristics. Only two studies were conducted in low- or middle-income countries, specifically China [60,61]. The remaining studies were conducted in high-income countries, with more than half of them conducted in the U.S. [62,63,64,65,66,67,68,69,70,71,72,73,74,75,76,77,78,79,80,81,82,83,84] or Australia [85,86,87,88,89,90,91,92,93,94,95,96,97,98,99]. Seven studies were carried out in the U.K. [100,101,102,103,104,105,106] and three each in The Netherlands [107,108,109] and New Zealand [110,111,112]. Two studies were conducted in both Germany [113,114] and Spain [115,116], while one study each was conducted in Canada [117], Denmark [118], Sweden [119], Portugal [120], Finland [121], and Poland [122]. We report key descriptive characteristics for all 63 studies in Appendix A. The quality of studies, as assessed by the CHEERS checklist, ranged from 52% to 100% for the included studies.

Twenty-six studies evaluated behavioural interventions [60,62,63,67,68,69,79,80,85,86,91,92,100,101,102,103,105,111,112,113,115,116,117,119,120,121]. Purely environmental [66,114] or policy [98,104] interventions were analysed in two studies each, while one study evaluated a surgical intervention [76]. Thirty-two studies analysed interventions that belonged to multiple categories [61,64,65,70,71,72,73,74,75,77,78,81,82,83,84,87,88,89,90,93,94,95,96,97,99,106,107,108,109,110,118,122]. Schools were the most common settings for intervention evaluation, with 23 interventions conducted in different educational settings [60,61,63,66,68,70,75,77,81,83,84,93,95,105,106,108,109,110,113,115,116,117,120]. Twenty-two interventions met the criteria for multi-setting implementation [64,71,73,74,78,79,80,82,87,88,89,90,91,94,98,99,100,101,111,112,114,122]. A total of five [69,76,107,119,121], two [72,92] and three [62,85,86] interventions were conducted in clinical, home, and family settings, respectively. Five interventions were targeted at community settings [67,96,102,103,118], while three study interventions took a population-wide perspective [65,97,104].

The underpinning study design was a solely trial-based economic evaluation for 27 studies [60,62,66,67,69,75,79,80,84,86,92,93,95,101,102,103,105,106,107,111,112,113,115,116,118,119,121]. Twenty-four studies involved decision-analytic modelling alone [71,72,73,74,77,78,81,83,87,88,89,90,91,94,96,97,98,100,104,114,117]. Six studies incorporated two types of study designs [61,68,70,76,99,109,110], which were predominantly a hybrid of trial-based and decision-analytic designs. The primary designs for the remaining studies were less common and incorporated cross-sectional [65], quasi-experimental [108], non-RCT [120], longitudinal [122], and pilot study [82] designs. Forty-five studies took a societal perspective [60,61,63,64,65,66,67,68,69,70,71,72,73,74,75,77,78,79,80,81,83,85,86,87,88,89,90,91,93,94,96,98,105,106,107,108,109,111,113,114,115,118,119,120,121], while 12 studies took a health care perspective [76,92,95,97,99,100,101,102,103,104,110,112].

The baseline study populations in the reviewed studies spanned all ages of childhood. The predominant study aim was to prevent excess weight gain, with interventions targeting solely healthy-weight children in 37 studies [60,61,63,65,66,68,70,71,72,73,74,75,77,81,83,84,88,89,90,91,92,93,94,95,96,97,98,99,104,106,110,111,113,114,115,117,119]. Only two studies took a management approach, where interventions offered some form of multidisciplinary supportive care [103,122]. Interventions in 14 studies took a treatment approach, where the baseline childhood populations had excess weight [62,67,69,76,79,82,85,86,100,101,102,112,118,121]. Of these studies, one was conducted in a cohort of children with weight-related comorbidities [112] and another study cohort involved adolescents with obesity and signs of addictive eating [82]. Interventions in 10 studies of children with mixed (normal and excess weight) body weight status took a combined approach of prevention and treatment [64,78,80,87,105,107,108,109,116,120]. Four studies were conducted in infant and pre-school populations [92,99,114,119]. Two studies targeted female-only cohorts [63,68], and the interventions in two other studies focused on children in low-income communities [84,93].

More than half of the studies were either cost-effectiveness analyses (CEAs) or CUAs. The most common measure of health consequence for the 30 CEAs in this review was change in BMI z-score or reduction in BMI units [60,62,65,66,67,69,70,79,84,85,87,88,89,90,91,92,93,94,95,96,98,100,101,112,113,116,118,119,121,122]. Other measures of health consequence included change in percentage body fat, waist circumference (WC), waist-to-height ratio (WHtR), number of obesity cases, percentage point change in the proportion of schools that adhered to the policy, minutes of moderate to vigorous physical activity (MVPA), and metabolic equivalent (MET). Eleven studies were exclusively CUAs with measures of consequences presented as QALYs, disability-adjusted life years (DALYs), or health adjusted life years (HALYs) [68,75,76,102,103,105,106,109,110,114,117]. Three studies were purely cost-benefit analyses (CBAs) with monetary valuation of consequences [81,83,108], two of which were school-based water interventions [81,83]. Eight studies were cost-consequences analyses, where disaggregated costs and outcomes, such as change in dietary habits, percentage of body fat, MVPA, and dietary habits (fruits/vegetables intake, water, desserts, and sugar-sweetened beverages (SSBs) served) were reported [78,80,82,86,97,104,115,120]. Eleven studies reported two or more types of economic evaluation [61,63,64,71,72,73,74,77,99,107,111].

There was a strong correlation between the time horizons of the economic evaluations and their study designs. Trial-based studies had follow-up periods ranging from 7 months [115] to 4 years [84,111]. Studies that featured decision-analytic modelling tended to have a lifetime horizon, although shorter time horizons were modelled in some studies. Discounting of future costs and consequences was not relevant to trial-based studies where the follow-up period did not exceed 1 year. Across most countries, the most frequently applied discount rate was 3%, but rates as high as 5% for both costs and consequences [92,99,111] and as low as 1.5% for consequences [107,109] were applied in some studies. A few studies with follow-ups and time horizons exceeding 1 year did not apply discount rates, and no justification was provided for the omissions [84,93,104,118,119,122].

Interventions in 50 of the reviewed studies were either cost-effective [60,62,63,64,65,75,85,87,91,96,99,107,113,114,116,117] or potentially cost-effective [61,66,67,68,69,70,71,72,73,74,76,77,78,79,80,81,82,83,84,92,93,94,95,97,98,100,101,104,109,110,111,112,115,118], while interventions in 13 studies were either not cost-effective [86,88,89,90,102,103] or inconclusive [105,106,108,119,120,121,122]. Some authors expressed concerns about the implementation of the intervention, even though they deemed the intervention cost-effective [71,74,95]. In relation to study approach, we estimated that 86%, 79%, and 70% of the interventions for prevention, treatment, and prevention/treatment were either cost-effective or potentially cost-effective, respectively. Neither of the two management interventions we reviewed were either cost-effective or potentially cost-effective. Other studies reported gender differences in the estimates of cost-effectiveness [80,106,116]. One study indicated that the intervention had shown greater health benefits and cost savings in those living in socioeconomically disadvantaged areas compared to those living in the least disadvantaged areas [94], and another study highlighted the uncertainties surrounding the persistence of effects beyond adolescence that could drive estimates of cost-effectiveness [109].

### 3.4. Economic Determinants and Broader Economic Consequences of Childhood Excess Weight

#### 3.4.1. Summary of Study Results—Economic Determinants

In this section, we focus on two key economic determinants of CEW, namely, socioeconomic inequalities, and food pricing and consumption. Our Google Scholar searches identified a relevant systematic review by El-Sayed et al. [123], the results of which form the core of our narrative review of socioeconomic inequalities in CEW. For food pricing and consumption, we cite findings from original studies identified by our searches.

##### Socioeconomic Inequalities

Socioeconomic status (SES), the measure of an individual’s combined economic and social status [124] or their proximity to resources relative to other individuals in society [123], is an established predictor of health outcomes. Many studies find that individuals in less favourable socioeconomic positions tend to present with poorer morbidity profiles and suffer higher mortality rates for chronic diseases [125]. Furthermore, there is evidence suggesting that childhood-to-adulthood BMI trajectories may be mediated by SES in some subgroups of the population [126].

Common measures of SES described in the literature can be broadly categorised as education-, income-, or occupation-based [124]. In the systematic review by El-Sayed and colleagues [123], SES metrics were defined at three levels: geographic- or area-level, household-level, and individual-level [123]. Most of the studies identified by this review that assessed geographic-level SES metrics using common indices such as the Townsend Deprivation Index [127,128] and the Carstairs–Morris Deprivation Index [129] found positive associations between the higher levels of area-level deprivation and obesity prevalence. For household and individual metrics, El-Sayed et al. [123] found mixed results. With the exception of a few contributing studies, metrics such as the head-of-household social class and maternal education were reliable predictors, showing inverse associations with childhood obesity. Metrics such as household income and receiving free school meals showed conflicting and weak associations, respectively, with childhood obesity across the reviewed studies. More recent studies and reviews of studies have echoed the mixed results for specific SES determinants identified by the systematic review by El-Sayed and colleagues [130,131,132,133].

##### Food Pricing

The 21st century has seen unprecedented access to an ever-increasing variety of food types across the world. Consequently, dietary behaviours have adapted to marked changes in food choice and availability. The evidence linking excess food consumption, often measured in terms of energy intake, to excess weight gain is compelling [134,135], especially for highly processed, energy-dense foods and sugar-sweetened beverages (SSBs). Food pricing is a relevant economic determinant because policies that affect food and beverage purchases are potential drivers of energy intake [136].

Several jurisdictions around the world have implemented population-wide policies aimed at improving food intake and reducing obesity, and there is increasing evidence of the effectiveness of SSB tax policies. A systematic review by Teng et al. [137] indicated that SSB taxes introduced in various geographical regions were effective in cutting SSB purchases. Recent research from the U.K. shows strong evidence that a soft drinks industry levy might limit exposure to liquid sugars and consequent health risks [138,139].

#### 3.4.2. Summary of Study Results—Broader Economic Consequences

We identified a systematic review by Segal et al. [9] published in 2020 that investigated the impact of CEW on human capital in high-income countries. We re-examined the 19 studies included in the systematic review as well as two additional studies identified by our searches [20,21] that reported labour market outcomes. A descriptive analysis of the individual studies is presented in Appendix A. The studies in this review were mostly U.S.-based, and the predominant datasets used were the U.S. National Longitudinal Survey of Youth 1997 (NLSY97), the National Longitudinal Study of Adolescent Health (Add Health), and the Early Childhood Longitudinal Study-Kindergarten Class Assessment (ECLS-K).

Cognitive performance was the most researched broader economic consequence, being the focus of 17 studies [140,141,142,143,144,145,146,147,148,149,150,151,152,153,154,155,156]. Four studies [20,142,154,157] focussed on educational attainment, while five studies from three articles [20,21,158] focused on labour market outcomes. Thirteen studies were based on data from the U.S. [20,140,141,142,143,144,148,150,152,153,154,157,158], while two studies each were based on data from the U.K. [146,155], Taiwan [145,149], and Australia [151,156]. One additional study used Canadian data [147] and another used Swedish data [21].

##### Cognitive Performance

The most common measure of cognitive performance was students’ standardised scores for academic subjects in national examinations, such as the Peabody Individual Achievement Test (PIAT), Key Stage examinations, and the National Assessment Program Literacy and Numeracy examination (NAPLAN). Cognitive performance tests also took the form of high school grade point averages (GPAs).

The analyses were stratified by sex in 13 studies [140,141,142,143,144,145,148,149,150,151,152,154,155], seven [140,141,143,148,149,151,154] of which reported that excess weight was associated with lower cognitive function. Two studies reported a significant negative effect of excess weight on cognitive performance for girls but not for boys [141,154]. However, Black et al. [151] found that obesity was negatively related to cognitive achievement for boys but not for girls. The results from the six studies [140,143,144,146,148,155] that performed disaggregated analysis by race were mixed with three [140,143,148] reporting lower cognitive function in association with excess weight and three [144,146,155] not reporting any significant effects. Only two studies were stratified by age: one reported no difference in cognitive function [142], while the other reported a greater magnitude of negative excess weight effects on senior year students compared with junior grade students [156].

##### Educational Attainment

In addition to three studies [142,154,157] identified by Segal and colleagues [9], our review includes a U.S.-based study by Amis et al. [20] that also examined educational attainment. Amis et al. [20] estimated the effects of being obese during adolescence on the likelihood of high school graduation and post-secondary educational attainment in the U.S., disaggregating their analysis by sex and race (whites, blacks, and Hispanics). Results were not significant for high school on-time graduation or college attendance for all children or any of the subgroups. However, children with obesity who went on to attend college were 9% less likely to graduate than their healthy-weight peers. The negative effects of being obese on college degree attainment were statistically significant overall as well as for female students and whites, who were both 12% less likely to graduate than nonobese adolescents.

Kaestner and Grossman [142] investigated the association between weight and grade attainment for children aged 9–10 and 11–12 years in the U.S., finding similar grades for overweight or obese children compared with children of normal weight in both age subgroups. Okunade et al. [157] examined the effects of weight on adolescents stratified by sex and race in the U.S. They found a statistically significant negative relationship between excess weight and timely high school completion in females but not in males. In terms of racial disparities, adverse effects were found primarily in white and Asian females, but no significant effects were reported for African Americans. Sabia and Rees [154] examined the effect of body weight on high school diploma and college completion among adolescents in the U.S. and found a stronger negative effect for female academic achievement relative to male academic achievement.

##### Labour Market Outcomes

In addition to the study [158] identified by Segal and colleagues [9], this review includes two articles [20,21] comprising four studies in the U.S., U.K., and Sweden. A U.S.-based study by Pinkston [158] examined the hourly wages of adults in 2009 who were overweight, obese, and severely obese aged 12 to 16 years in 1996 (white children only). It found that a childhood history of severe obesity had a significant large negative effect on wages for white men, whilst in women, a childhood history of both overweight and obesity had a significant and large negative effect. Furthermore, an initial penalty of 13% in wages was estimated for women entering the labour market with a BMI over 37.

A U.S.-based study by Amis and colleagues [20] followed white, black, and Hispanic children of both sexes with a mean age of 16 years for 13 years and estimated labour market earnings. The authors found that the negative effects of adolescent obesity on future earnings were predominant among females and blacks. They earned nearly 12% and 9% less, respectively, in future wages, compared to their non-obese peers.

Lundborg and colleagues [21] estimated the Swedish adult male labour market penalty for being overweight or obese as a teenager and carried out a complementary analysis on cohorts of children in the U.S. and the U.K. In their large sample of 145,193 siblings who enlisted for the military at age 18, they found that, in comparison with teenagers of normal weight, overweight and obese adolescents earned 6% and 18% less, respectively, between ages 28 and 38. Both estimates were statistically significant. To put these estimates in context, they calculated that the penalty of being obese corresponded to almost three years of schooling, or the time required to complete a university bachelor’s degree. Using the United Kingdom National Child Development Study (NCDS) cohort, the authors further estimated that being obese at 16 was associated with a statistically significant 38% decrement in earnings at age 42. Estimates for overweight at the same age were substantially lower (about 2%) and not statistically significant. Using the NLSY79 data, Lundborg and colleagues estimated that men who were obese at ages 16 to 24 were likely to earn 18% less at ages 39 to 42 relative to their normal-weight counterparts.

## 4. Discussion

### 4.1. Economic Costs Associated with Childhood Excess Weight

#### 4.1.1. Summary of Results and Comparative Evidence

Our review identified six additional studies since the systematic review by Hamilton et al. [13], some of which applied new statistical methods for the estimation of economic costs associated with CEW. We demonstrated that childhood overweight and obesity lead to significant short- and long-term excess direct and indirect costs in high-income countries. Furthermore, this review revealed a number of gaps in the evidence base in this area. First, most of the studies were conducted in the U.S., with the remaining conducted in Australia, Germany, and The Netherlands. There is a paucity of evidence surrounding the economic costs of CEW in low- and middle-income countries. Second, only two studies estimated indirect costs, and third, there was lack of clarity regarding the methodological approaches adopted by many of the reviewed studies.

The studies included in our review estimated that, in high-income countries, the mean excess annual direct costs of overweight and/or obesity per child ranged between GBP 89 and GBP 1829 (pounds sterling, 2020 prices). Over a lifetime horizon, excess direct costs per child ranged from GBP 1494 to GBP 13,260 (pounds sterling, 2020 prices) in the reviewed studies. We note that the annual estimates are not proportional to lifetime estimates. Van Baal et al. [19] argue that excess costs due to diseases unrelated to obesity in adulthood could potentially offset costs from obesity prevention. It is possible that the excess costs due to overweight and obesity exhibit a non-linear increase from childhood to old age. These ranges should be considered with caution, considering the substantive methodological differences across the reviewed studies.

Only two studies investigated indirect costs, limiting the usefulness of a presentation of ranges in values. It has been suggested that indirect costs account for a greater share of total economic costs compared to direct costs [159], but the current literature on CEW is insufficient to establish this. The lifetime indirect cost projections by Sonntag et al. [33] show significantly lower costs for girls compared to boys. They argue that this difference is due to lower rates of full-time employment and lower lifetime earnings among women during adulthood.

Meta-analyses of economic costs, as attempted in the systematic review by Hamilton et al. [13], are unlikely to yield estimates that are generalisable across different jurisdictions and countries due to the high level of between-study heterogeneity stemming from the type of outcome estimated, as well as variations in health care practices and relative prices of resource inputs across settings. Charges and expenditures reported in most U.S.-based studies have been shown to differ considerably for a given procedure [22]. Further inter-study variability stems from the body weight status categories compared, the population subgroups investigated, and the type of costs included in the analysis of individual studies. The global mean total lifetime excess cost calculated in the systematic review of 13 studies by Hamilton et al. [13] for boys (GBP 142,258—pounds sterling, 2020 prices) is an extreme departure from the estimate by Sonntag et al. [32,33] (GBP 8958—pounds sterling, 2020 prices) for males in the German population and demonstrates the challenge associated with synthesising costs outcomes across studies.

Two studies that examined the impact of CEW on health care utilisation using an IV estimator found elevated utilisation for excess weight relative to healthy weight when endogeneity was accounted for [160,161]. Among the six regression-based studies we identified, two studies conducted in Australia [36] and the U.S. [37] applied an IV estimator to mitigate omitted variable bias. In both studies, IV estimates of economic burden were significantly larger than non-IV estimates within the studies as well as earlier and contemporary non-IV estimates from other studies. The authors suggested that measurement error in children’s BMI and unobserved variables could be responsible for the underestimated economic cost outcomes from non-IV estimates. We note, however, that the IV estimator is not without limitations. Primarily, when only one instrument is used—as in the reviewed studies—its validity cannot be fully established.

#### 4.1.2. Strengths and Limitations

The main strength of our study is the review methodology, which entailed the presentation of costs in a common currency and price date, thereby aiding comparisons across study estimates, and the use of narrative and tabular synthesis methods. The major limitation to our approach is that only one database was searched, and some studies of economic costs may therefore have been missed.

Our quality assessment of the studies using the CHEERS checklist identified several areas for improvement in studies aiming to estimate the economic burden of CEW. Generally, these include applying a discount rate when the study time horizon exceeds one year, performing robust deterministic and probabilistic sensitivity analyses where appropriate, accounting for the variation in results across subgroups of patients where possible, reporting sources of funding, and describing conflicts of interest. Future studies in this strand specific to a childhood populations should consider Mendelian randomisation, an IV method that uses selected genetic variants [162].

#### 4.1.3. Research and Policy Implications

Econometric/regression analysis and decision-analytic modelling approaches for estimating the economic burden of CEW have practical applications for health care planning and health policy impact assessment. For instance, decision makers may be interested in estimating the impact on economic costs of hypothetical reductions in the prevalence of childhood obesity [33] or the potential lifetime cost savings accrued from scenarios of varying intervention costs and effect sizes [100]. Future studies that address the knowledge gaps identified in this review should further contribute to the literature on the economic burden of CEW.

### 4.2. HSUVs Associated with Childhood Excess Weight

#### 4.2.1. Summary of Results and Comparative Evidence

Similar to the systematic review by Brown and colleagues [14], our review revealed a dearth of evidence on HSUVs for CEW with 14 included studies from only four high-income countries. Several methodological challenges characterised chiefly by considerable variation in valuation protocol (utility instrument, population preferences for weight-related health states, and methods of valuation) and sample sizes (76 children [44] to 4979 children [47]) across studies may have impacted the quality of the estimated utilities. Inadequately powered sample sizes, the lack of sensitivity of the instruments used in children of young ages, or the weak relationship between BMI and utility could potentially have diminished the statistical significance of estimated utility values associated with a given weight status [49,51,53].

In contrast to the studies from the original review by Brown et al. [14], which were all cross-sectional studies, three of the newly identified studies from our searches were based on longitudinal data analysis [52,54,56]. Longitudinal study designs offer an opportunity to investigate reverse causation [14], and these recent longitudinal studies suggest a growing interest in investigating theories about the potential relationships between childhood body weight status and HSUVs.

A recent systematic review that investigated the longitudinal association between weight change and health-related quality of life (HRQoL) in children did not include studies that assessed HSUVs from generic preference-based measures [163], and preference-based measures are often omitted in many clinical effectiveness studies on childhood weight gain. This exclusion underlines the need for mapping functions that predict HSUVs from a target preference-based measure using data obtained from non-preference-based HRQoL measures. Mapping functions or prediction equations are increasingly being developed for utility instruments. One example is the mapping of a weight measure called the Weight-Specific Adolescent Instrument for Economic Evaluation (WAItE) to the CHU-9D [164]. It is worth noting that mapped estimates of utilities are subject to greater uncertainty, and mapping may generate HSUVs with poor external validity [165]. Therefore, it is preferable to include utility instruments directly in RCTs, and where a preference-based measure is not included, the appropriateness of a mapping function should be considered rigorously.

The quality assessment of individual studies included in this updated review used the key criteria recommended by Papaioannou et al. [25] and represents an important step for assessing the rigour and reliability of the reviewed studies. However, further quality considerations may be worth exploring. For instance, in addition to the question around the appropriateness of the measure, it may be worth considering if the population that valued the change in HRQoL is appropriate. A points-based checklist that gives additional weight to longitudinal studies may also add more rigour to the quality assessment procedure.

#### 4.2.2. Strengths and Limitations

While our study draws from a growing body of longitudinal studies and highlights the need for a fine-tuned quality assessment tool, we note that it has weaknesses. One limitation of our study is that only one database was searched and relevant studies on preference-based measures in CEW may have been missed. We also did not carry out an updated meta-analysis. Given the limited scope of our search, such an analysis may exclude key studies. Therefore, we recommend an exhaustive search and updated systematic review of this topic.

#### 4.2.3. Research and Policy Implications

HSUVs are crucial elements of CUA, which is the method recommended for carrying out economic evaluations for health care decision making by the National Institute for Health and Care Excellence (NICE), the Pharmaceutical Benefits Advisory Committee (PBAC), and several other health technologies regulatory bodies around the world [166,167,168]. The methodological choices of primary studies, as well as mapping studies estimating HSUVs associated with CEW, can introduce significant uncertainty to CUAs. Therefore, careful consideration should be given to sources of heterogeneity, such as the valuation protocol and sample size when conducting or assessing studies of HSUVs in this area.

### 4.3. Economic Evaluations of Interventions Targeting Childhood Excess Weight

#### 4.3.1. Summary of Results and Comparative Evidence

This review identified emerging trends around economic evaluations of interventions for children with excess weight. Of note is the sharp increase in recently published economic evaluations, with nearly half of the identified studies conducted within the past three years. This is indicative of growing public and decision maker interest in the trade-offs between the costs and consequences of competing interventions. Although we estimated a higher percentage of preventive interventions that were either cost-effective or potentially cost-effective compared with treatment interventions (86% versus 79%), in general, there is still uncertainty about which strategy is likely to be more cost-effective, as most of the studies we reviewed focussed on preventive interventions. However, we note a shift towards economic evaluations of treatment interventions in recent years, with only nine solely treatment interventions between 2001 and 2015 versus five between 2016 and 2020.

The most frequently adopted approach to economic evaluation in this review was CEA, where measures of consequences were typically captured in terms of units of change in body weight status such as change in BMI z-score. Despite increasing adoption in other areas of health care, CUA remains a less common approach in this field. The use of CUA, particularly in trial-based study designs, is limited by the methodological challenges arising from the measurement and valuation of health states in childhood populations discussed in the utilities section above (Section 4.2). CUA was the main method of choice for decision-analytic models adopting a longer-term time horizon because utility estimates for health states in adulthood are more established. However, longer-term modelling of costs and consequences introduces uncertainties due to assumptions about the persistence of benefits accrued from interventions implemented over short periods during childhood.

While the systematic review by Zanganeh et al. [15] did not identify cost-benefit analyses, this review identified three economic evaluations that applied a purely cost-benefit approach [81,83,108]. Cost-benefit analyses of interventions targeting CEW tend to be more commonly conducted in the U.S. In the U.K. and Australia, CBA is usually recommended for the evaluation of public sector interventions outside the health sector. The differences in approaches to economic evaluations across countries hinder comparability. Torbica et al. [169] argue that a ‘one size fits all’ approach is not feasible, and the chosen evaluative method will inevitably reflect contexts unique to individual countries. Because most childhood obesity preventive interventions are implemented outside the health sector, the inclusion of cost-benefit analyses may give further insights in terms of wider societal budgeting and planning. For instance, Moodie et al. [90] found that although the TravelSMART Schools Curriculum programme was not cost-effective following a cost-utility-based analysis, the intervention could be cost-effective if non-obesity consequences such as environmental outcomes are considered.

There are indications that interventions may be more cost-effective when implemented in subgroups of the population. For instance, two studies [80,116] found that certain school-based interventions were cost-effective in boys but not in girls, while another study [106] reported an intervention to be cost-effective in girls but inconclusive in boys. The mechanisms to explain these potential differences are unclear. Future studies should strive to estimate and explain any differences in cost-effectiveness that may exist between key population subgroups.

#### 4.3.2. Strengths and Limitations

A key strength of this review is the inclusion of intervention categories excluded from the previous review [15]. One evaluation of a surgical intervention was identified and showed potentially cost-effective estimates for bariatric surgery in children with severe obesity. Future reviews should widen the search for novel interventions such as pharmacological and technology-based interventions such as eHealth/mHealth interventions. As with the other reviews in this study, we carried out our search in only one database, and relevant studies might have been missed. However, the number of newly published studies identified by our search was nearly as many as those identified in the systematic review that we updated and augmented.

Future studies in this area should account for differential timing of outcomes when the study time horizon exceeds one year. Robust deterministic and probabilistic sensitivity analyses should be performed where appropriate, and the variation in results across subgroups of patients should be accounted for. For studies that project cost-effectiveness beyond the period a follow-up period, extensive scenario analyses should be conducted to test assumptions about the persistence of treatment effects. Finally, the sources of funding and conflicts of interest should be stated by authors.

#### 4.3.3. Research and Policy Implications

The objective of economic evaluation is to provide guidance for the choice of interventions targeting CEW. Ideally, the type of economic evaluation applied should reflect the analytical perspective and the institutional context of the geographical jurisdiction. However, evaluating interventions targeting CEW is distinctively challenging because they tend to comprise multiple interacting components with implementation often outside the health sector [170]. A further complication arises from the uncertainty surrounding the assumptions of longer-term costs and consequences of interventions in early childhood. Economic modellers argue that these assumptions are necessary, since all the perceived benefits and cost offsets may not be apparent within the timeframe of intervention implementation [171], but the choice of longer-term costs and consequences can significantly influence investment decisions.

### 4.4. Economic Determinants and Broader Economic Consequences of Childhood Excess Weight

#### 4.4.1. Summary of Results and Comparative Evidence

The evidence identified by our review of economic determinants and broader economic consequences of CEW shows mixed or weak associations for some SES metrics such as household income and receipt of free school meals. The underlying mechanisms of association between SES measures and CEW are not fully understood [123]. Longitudinal studies could potentially disentangle causal relationships in this area.

Economists and policymakers argue that the excess obesity-related costs imposed on society justify government interventions to impose a levy on nutritionally poor foods and/or subsidise healthier choices [172]. Theoretically, an increase in the price of an energy-dense food should lead to less demand for such food [137]. Emerging studies in the U.K. show large population-wide health benefits from implementing a soft drinks industry levy, though this is mediated primarily through reformulation of products rather than changes in purchasing behaviour [138,139]. Non-price mechanisms such as the health signalling pathway where even small taxes influence voluntary behaviour change in the public, reduced portion sizes, and product reformulation have also been posited [137,173]. However, the likelihood of compensatory behaviour where SSBs are substituted for other untaxed energy-dense foods could present additional challenges [174].

A negative association between childhood exposure to excess weight and cognitive performance for girls was re-affirmed in this review, but it is debatable if there is an age gradient with respect to the effect of CEW on cognitive performance. In sum, the generalisability of the evidence on human capital outcomes is constrained by the predominance of studies from the U.S. and studies focused on cognitive outcomes.

#### 4.4.2. Strengths and Limitations

This review provides evidence of the negative impact of CEW on human capital outcomes by including four studies on educational attainment and labour market outcomes. Our searches were, however, limited for this strand of research.

#### 4.4.3. Research and Policy Implications

The priority for future research should focus on explaining the pathways that mediate the above findings. The evidence on variations in outcomes by ethnicity, SES, and other subgroups is also limited. This suggests the need for more studies with disaggregated analyses.

## 5. Conclusions

This study represents a concise, structured review of economic aspects of CEW conducted by reviewing evidence across four research strands. We highlighted knowledge gaps in all the research strands. The reviewed studies on economic costs establish a positive association between increasing CEW and economic costs. However, establishing a central estimate is hindered by considerable heterogeneity in the literature. Across studies, the mean annual excess direct costs associated with CEW ranged from GBP 89 and GBP 1829, whilst the mean lifetime excess direct costs ranged from GBP 1494 to GBP 13,260 (2020 prices). In the context of health economic evaluation, preventive interventions have been the primary focus and the most likely to be demonstrated to be cost-effective, although recent years have seen a notable shift towards the economic evaluation of treatment approaches. Evidence from this structured review can act as data input into future economic evaluations in this area and highlights areas where future economic research should be targeted.

## Figures and Tables

**Table 1 children-09-00461-t001:** Aggregate descriptive summary of economic cost studies.

Study Characteristics	Number of Studies Identified
Year of publication	
2006–2010	3 [28,29,30]
2011–2015	2 [31,32]
2016–2020	7 [33,34,35,36,37,38,39]
**Country/Jurisdiction**	
High-income	All
Low- and middle-income	None
**Type of study design (or decision-analytic model, if applicable)**	
Decision-analytical models	2 [31,34,35,36,37,38]
Longitudinal study/panel data analysis	6 [28,29,30,32,33,39]
**Study perspective**	
Direct costs	11 [28,29,30,31,32,34,35,36,37,38,39]
Indirect costs	2 [29,33]
**Study time horizon**	
Lifetime	5 [28,30,32,33,39]
30 years	1 [29]
10 years	1 [38]
3 years	1 [34]
1 year	4 [31,35,36,37]

**Table 3 children-09-00461-t003:** Descriptive analysis of the key items on health utility values associated with childhood excess weight.

First Author and Year of Publication	Country	Population and Age	Sample Size(s)	Type of Study	Utility Instrument (Proxy Assessment)	Preference/Valuation Method	Estimated Utility Values (or Coefficients) and Corresponding Health States	Quality Score (%)
Boyle, S.E., et al. 2010 [43]	United Kingdom	Children aged 11 to 15 years: two groups who either achieved the recommended PA guidelines or did not	*n* = 1771 (achieved recommended PA: Yes *n* = 446; No *n* = 1325)	Cross-sectional study	EQ-5D-Y/VAS	United Kingdom adult general population/TTO	Healthy/normal weight: 0.9 (s.d. 0.18); Overweight or obese: 0.87 (s.d. 0.14)	4.5/6 (75%)
Belfort, M.B., et al. 2011 [44]	United States	Children aged 5 to 18 years	*n* = 76 (Healthy weight *n* = 34; Overweight or obese *n* = 42)	Cross-sectional study	HUI3 (and proxy parent version for all)	Canadian general population (>16 years of age)/SG	Healthy weight: 0.81 (95% CI 0.76–0.86); Overweight or obese: 0.78 (95% CI 0.72–0.83)	4.5/6 (75%)
Keating, C.L., et al. 2011 [45]	Australia	Secondary school children aged 12 to 15 years	*n* = 2890 (Thin *n* = 16; Healthy weight *n* = 1960; Overweight *n* = 642; Obese *n* = 272)	Cross-sectional study	AQoL-6D	Recalibrated for Australian adolescents/TTO	Healthy/normal weight: 0.86 (s.d. 0.16); Overweight: 0.842 (s.d. 0.17); Obese: 0.805 (s.d. 0.18)	4.5/6 (75%)
Makkes, S., et al. 2013 [46]	The Netherlands	Children aged 8 to 13 years and 13 to 19 with severe obesity	8 to 13 years old (*n* = 16) and 13 to 19 years old (*n* = 64)	Cross-sectional estimations from the HELIOS trial	EQ-5D-3L VAS	Dutch general population/TTO	Severely obese: 0.79 (s.d. 0.22)	4/6 (67%)
Trevino, R.P., et al. 2013 [47]	United States	Sixth grade students (aged under 13 years and approximate average age 11 years)	*n* = 4979 (BMI%: <85 *n* = 2456; 85–94 *n* = 1003; 95–99 *n* = 1176; and 99+ *n* = 344)	Cross-sectional estimations from the HEALTHY trial	HUI2 and HUI3	Canadian general population (>16 years of age)/SG	HUI2 BMI%: <85 0.853 (s.d. 0.157); 85–94 0.848 (s.d. 0.157); 95–99 0.838 (s.d. 0.163); and 99 + 0.814 (s.d. 0.175) and HUI3 BMI%: <85 0.805 (s.d. 0.233); 85–94 0.795 (s.d. 0.236); 95–99 0.786 (s.d. 0.242); and 99 + 0.759 (s.d. 0.245)	6/6 (100%)
Bolton, K., et al. 2014 [48]	Australia	Students aged 11 to 19.6 years	*n* = 1583 (Healthy weight *n* = 727 and Overweight/obese *n* = 243)	Cross-sectional study (baseline data only)	AQoL-6D	Recalibrated for Australian adolescents/TTO	Healthy/normal weight: 0.89 (s.d. 0.14) and Overweight or obese: 0.87 (s.d. 0.14)	4.5/6 (75%)
Canaway, A. and E. Frew 2014 [49]	United Kingdom	Children aged 6 to 7 years	*n* = 160 (Normal weight *n* = 127; Overweight *n* = 13; Obese *n* = 20; and Overweight or obese *n* = 330	Cross-sectional study	CHU-9D and EQ-5D-Y	United Kingdom adult general population/CHU-9D: SG and EQ-5D-Y: TTO	CHU-9D: Healthy/normal weight 0.87 (95% CI 0.84–0.89); Overweight 0.86 (95% CI 0.81–0.9); Obese 0.84 (95% CI 0.77–0.91); and Overweight or obese 0.85 (95% CI 0.8–0.89) and EQ-5D-Y: Healthy/normal weight 0.73 (95% CI 0.66–0.8); Overweight 0.66 (95% CI 0.43–0.83); Obese 0.69 (95% CI 0.54–0.83); and Overweight or obese 0.67 (95% CI 0.56–0.78)	4.5/6 (75%)
Chen, G., et al. 2014 [50]	Australia	Primary (7 to 13 years) and secondary (13 to 17) school children	Primary schools *n* = 2588 (Healthy-weight *n* = 1674; Overweight *n* = 396; Obese *n* = 107) and secondary schools *n* = 765 (Healthy-weight *n* = 520; and Overweight or obese *n* = 101)	Cross-sectional study (baseline data only)	CHU-9D	Recalibrated for Australian adolescents/SG	Primary schools: Healthy-weight 0.87 (s.d. 0.11); Overweight 0.86 (s.d. 0.12); Obese 0.83 (s.d. 0.16) and secondary schools: Healthy-weight 0.82 (s.d. 0.12); and Overweight or obese 0.81 (s.d. 0.12)	5/6 (83%)
Frew, E.J., et al. 2015 [51]	United Kingdom	Children aged 5 to 6 years	*n* = 1344 (Healthy weight 1012; Overweight 116; Obese 176)	Cross-sectional estimations from the WAVES trial	CHU-9D	United Kingdom adult general population/SG	Healthy weight 0.825 (s.d. 0.14); Overweight 0.811 (s.d. 0.14); Obese 0.827 (s.d. 0.13); and Overweight or obese 0.82 (s.d. 0.13)	4.5/6 (75%)
Eminson, K., et al. 2018 [52]	United Kingdom	Children between 6 and 10 years old	The WAVES trial: 1350 children at baseline (Healthy weight *n* = 1022; Overweight *n* = 118; Obese *n* = 167)	Longitudinal study	CHU-9D	United Kingdom adult general population/SG	In the regression results from the analyses investigating the impact of weight status on health utility, the coefficients (*p*-values) for healthy weight, overweight, and obese are 0.000437 (0.968), −0.00126 (0.915), and 0.003166 (0.782), respectively	6/6 (100%)
Tan, E.J., et al. 2018 [53]	Australia	Children aged 5 years	*n* = 368 (Healthy weight *n* = 224; Overweight *n* = 114; Obese *n* = 30)	Longitudinal study, but HRQoL data and analysis were cross-sectional	HUI3 (parent proxy version)	Canadian general population (>16 years of age)/SG	Healthy weight: 0.956 (*p* value 0.08); Overweight 0.956 (*p* value 0.09); Obese 0.952 (0.10). Utility estimates across the 3 weight status groups were similar.	6/6 (100%)
Hoedjes, M., et al. 2018 [54]	The Netherlands	Children and adolescents ages 8 to 19 years with severe obesity: intensive lifestyle treatment	*n* = 120	Longitudinal study	EQ-5D-3L	Dutch general population/TTO	Utility scores at baseline, after 1 year of treatment, and 1 year of follow-up were 0.80 (*p*-value 0.02), 0.89 (*p*-value 0.02) and 0.88 (*p*-value 0.02), respectively.	5.5/7 (79%)
Bell, L., et al. 2019 [55]	Australian	9–11-year-olds	*n* = 2611 at baseline (intervention *n* = 1373; 20 matched comparison *n* = 1238)	Quasi-experimental repeat cross-sectional design	CHU-9D	Recalibrated for Australian adolescents/SG	Utility values for the intervention at baseline and end of study were 0.82 and 0.77, respectively. Utility values for the comparator at baseline and end of study were 0.80 and 0.79, respectively. Utility values not reported by weight status.	7/7 (100%)
Killedar, A., et al. 2019 [56]	Australian	Two cohorts (waves 6 and 7) of boys and girls 10–17 years from the LSAC study	Girls: between *n* = 1370 and *n* = 1714 across cohorts/waves. Boys: between *n* = 1464 and *n* = 1778 across cohorts/waves	Primary data collection from a longitudinal study	CHU-9D	The best–worst scaling study conducted in an Australian adolescent population/SG	Girls: BMI z-scores −2, 1, 2, and 3 from ages 10 to 17, respectively: [10 years: 0.818; 0.812; 0.809; 0.807], [11 years: 0.814; 0.799; 0.794; 0.789], [12 years: 0.811; 0.787; 0.779; 0.771], [13 years: 0.807; 0.775; 0.764; 0.753], [14 years: 0.804; 0.762; 0.748; 0.735], [15 years: 0.800; 0.750; 0.733; 0.717], [16 years: 0.796; 0.738; 0.718; 0.698], [17 years: 0.793; 0.725; 0.703; 0.680]. Boys: BMI z-scores −2, 1, 2, and 3 from ages 10 to 17, respectively: [10 years: 0.811; 0.799; 0.795; 0.792], [11 years: 0.817; 0.806; 0.802; 0.798], [12 years: 0.824; 0.812; 0.809; 0.805], [13 years: 0.830; 0.819; 0.815; 0.811], [14 years: 0.837; 0.825; 0.822; 0.818], [15 years: 0.843; 0.832; 0.828; 0.824], [16 years: 0.850; 0.838; 0.835; 0.831], [17 years: 0.856; 0.845; 0.841; 0.837]	7/7 (100%)

AQoL: Assessment of Quality of Life; BMI: body mass index; CHU-9D: Child Health Utility-9 dimensions; EQ-5D: EuroQol-5 dimensions; EQ-5D-Y: EQ-5D-Youth; EQ-5D-3L: EQ-5D-3 levels; HUI2 and HUI3: Health Utilities Index version 2 and 3; LSAC: Longitudinal Study of Australian Children SG: standard gamble; TTO: time trade-off; VAS: visual analogue scale.

**Table 4 children-09-00461-t004:** Aggregate descriptive summary of economic evaluation studies.

Study Characteristics	Number of Studies Identified
Year of publication	
2001–2005	2 [62,63]
2006–2010	11 [64,65,66,67,85,86,87,88,89,111,121]
2011–2015	16 [60,68,69,70,71,72,73,74,90,91,92,100,101,110,113,115]
2016–2020	34 [61,75,76,77,78,79,80,81,82,83,84,93,94,95,96,97,98,99,102,103,104,105,106,107,108,109,112,114,116,117,118,119,120,122]
**Jurisdiction**	
High-income	61 [62,63,64,65,66,67,68,69,70,71,72,73,74,75,76,77,78,79,80,81,82,83,84,85,86,87,88,89,90,91,92,93,94,95,96,97,98,99,100,101,102,103,104,105,106,107,108,109,110,111,112,113,114,115,116,117,118,119,120,121,122]
Low- and middle-income	2 [60,61]
**Intervention category**	
Behavioural	26 [60,62,63,67,68,69,79,80,85,86,91,92,100,101,102,103,105,111,112,113,115,116,117,119,120,121]
Environmental	2 [66,114]
Policy	2 [98,104]
Surgical	1 [76]
Multiple categories	32 [61,64,65,70,71,72,73,74,75,77,78,81,82,83,84,87,88,89,90,93,94,95,96,97,99,106,107,108,109,110,118,122]
**Study approach**	
Prevention	37 [60,61,63,65,66,68,70,71,72,73,74,75,77,81,83,84,88,89,90,91,92,93,94,95,96,97,98,99,104,106,110,111,113,114,115,117,119]
Treatment	14 [62,67,69,76,79,82,85,86,100,101,102,112,118,121]
Treatment and prevention	10 [64,78,80,87,105,107,108,109,116,120]
Management	2 [103,122]
**Setting**	
School-based	23 [60,61,63,66,68,70,75,77,81,83,84,93,95,105,106,108,109,110,113,115,116,117,120]
Health care/clinical setting	5 [69,76,107,119,121]
Family	3 [62,85,86]
Home	2 [72,92]
Community	5 [67,96,102,103,118]
Population	3 [65,97,104]
Multi-setting	22 [64,71,73,74,78,79,80,82,87,88,89,90,91,94,98,99,100,101,111,112,114,122]
**Study design**	
Randomised controlled trial	27 [60,62,66,67,69,75,79,80,84,86,92,93,95,101,102,103,105,106,107,111,112,113,115,116,118,119,121]
Decision-analytical	24 [63,64,71,72,73,74,77,78,81,83,85,87,88,89,90,91,94,96,97,98,100,104,114,117]
Multiple design (studies with two main types of designs)	7 [61,68,70,76,99,109,110]
Cross-sectional	1 [65]
Quasi-experimental	1 [108]
Non-randomised controlled trial	1 [120]
Longitudinal	1 [122]
Pilot	1 [82]
**Study Perspective**	
Societal	45 [60,61,63,64,65,66,67,68,69,70,71,72,73,74,75,77,78,79,80,81,83,85,86,87,88,89,90,91,93,94,96,98,105,106,107,108,109,111,113,114,115,118,119,120,121]
Health care	12 [76,92,95,97,99,100,101,102,103,104,110,112]
Institutional or school system	2 [116,117]
Provider	1 [82]
Not stated/insufficient information	3 [62,84,122]
**Type of economic evaluation**	
Cost-effectiveness	30 [60,62,65,66,67,69,70,79,84,85,87,88,89,90,91,92,93,94,95,96,98,100,101,112,113,116,118,119,121,122]
Cost-utility	11 [68,75,76,102,103,105,106,109,110,114,117]
Cost-consequence	8 [78,80,82,86,97,104,115,120]
Cost-benefit	3 [81,83,108]
Two or more types	11 [61,63,64,71,72,73,74,77,99,107,111]

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
