# Peer review of "Health Economic Aspects of Childhood Excess Weight: A Structured Review"

_children, 2022, doi:10.3390/children9040461_

Round 1
Reviewer 1 Report
Referee report on Health economic aspects of childhood excess weight: a structured review By Olu Onyimadu, Mara Violato, Nerys Astbury, Susan A. Jebb, and Savros Petrou Date: 16 Feb. 2022 A. General remarks 1. The principal contribution of this paper is to collect recent economic evidence on four topics: (1) economic costs associated with childhood excess weight (CEW); (2) associated health utilities with CEW; (3) economic evaluation of interventions targeting CEW; and (4) economic determinants and broader consequences of CEW. The authors are to be commended for their massive research effort. 2. An important weakness of the paper is the lack of clear conclusions. Readers tend to be lost in the details. E.g. it would be helpful to have a central estimate of the costs in strand (1), or even better, the most credible estimate according to the authors. 3. Also, a comparison with other public health issues (e.g. tobacco) would place the findings in perspective. 4. Another weakness is the paper´s style in several places (see detailed comments below). B. Detailed comments (line) Section 1
- “Our ‘de novo’ searches identified six additional studies ”. In what sense additional? Since some year?
- “childhood excess weight”. Maybe worthwhile to introduce the acronym CEW.
- “impose substantial health care and indirect costs on society”. At this point, it is crucial to distinguish costs that are borne by the children (and parents) themselves and external costs borne by society. From an economic point of view, only the latter may (not automatically do!) justify public intervention. A difficulty is costs that parents impose on their children. Using them for justifying public intervention within the family is a slippery slope however because parents burden their children with all sorts of cost (but benefits, too, which may suffer from external intervention). Some pertinent language would be valuable (without necessarily passing a judgment).
53. “The first, a systematic review by”. “The first, by” is sufficient. 54. “research strand 1)”. At this point, the four strands have not been identified in the body of the paper. Pertinent language needs to be moved up from line 93f.
- “the study was constrained by only including studies that estimated costs over a lifetime horizon”. What is wrong with an estimate covering a person´s lifetime? Conversely, readers should be told the rate of discounting used in this study.
- “stated by the Hamilton and colleagues”. Delete “the”.
- “longitudinal study designs, which are crucial for exploring potential reverse causation.” For some readers at least, it would be helpful to offer more explanation. Also, the direction of causation may still not be determinable even through panel data as soon as the dependent variable (productivity) and the explanatory one (CEW) contain error terms that are correlated over time. An example would be parents´ low education.
- “Most of the studies”. “Most of them” would avoid repetition.
- “The systematic review by Zanganeh et al. [14] excluded pharmacological and surgical interventions.” Again, “This review excluded pharmacological and surgical interventions” is sufficient.
- “Segal et al. [9] omitted“. “The authors omitted” would be nicer.
Section 2
- “we provide our methods for updating”. Authors are unlikely to provide methods. Either write, “information about our methods” or “document our methods”.
- “The previous review”. Write, “previously cited review”
- “used by Hamilton et al. [12]”. Again, repetitive. Write, “by the authors”.
- “the age upon study entry”. Preferable to write, “age at study entry”.
- “in the English language.”. “in English” is sufficient.
- by Zanganeh and colleagues [14]”.
- “CHEERS statement”. Has this acronym been explained?
- “Evidence on the cost-effectiveness of an intervention”. “Evidence” is sufficient in this context.
Section 3
- “of these papers”. “them” is sufficient – English has pronouns!
- “or language of full text publication”. Presumably, “in the full text publication”.
- “one year to a lifetime”. Write, one year to lifetime”
- “The age ranges upon study entry captured children between”. This statement is not clear. Maybe, “Age at study entry varied between 2 and 18 years”?
- “However, both approaches are umbrella terms. “However, these are umbrella terms” is sufficient.
- Decision-analytic modelling”. “modelling” can be deleted.
- Three US-based studies [29,30,36] [38] reported various types”. This looks like four studies.
Table 2. “Currency unit (Price year)”. Better two write, “Currency (at prices of the year x)”
- “Mean individual-level excess lifetime costs relative to healthy weight ranged from £1,494”. It is puzzling that nowhere is the cost per kg excess weight discussed. Should it not make a difference whether a person has 5 kg or 10 kg excess weight?
- “Lightwood et al. [28] (2020 price year).” Write, “(at 2020 prices)”.
- “estimated” – six times in one para. -- a bit more variation would be nice.
- “The age ranges upon study entry captured children between 3-18 years.” Unclear statement. Age does not “capture” anyone.
- “Sonntag et al. [32] estimated excess lifetime indirect costs due to excess weight per girl and boy at £2,586 and £4,451, respectively (2020 price year). While for boys direct and indirect costs estimated by the same authors in related studies were similar, for girls, the direct excess costs (£7,432) were nearly three times the indirect costs (2020 price year) in the German population.” Preferably this should follow directly after the first mention of the study on line 1. Also, are the low relative indirect costs for girls caused by their lower lifetime earnings?
- “The highest estimates of utilities for healthy weight”. This statement is ambiguous, it might relate to an extrapolation from actual to ideal weight. Better to write, “associated with healthy weight in the sample”.
- “They found that the association between BMI z-score and utilities was affected by age and was statistically significant in girls but the association was weak in boys at all ages”. Somewhat amazingly, the “raw differences” in utility are small. Adding a sentence or two stating that small differences may conceal substantial ceteris paribus effects of excess weight would be useful.
- “32 studies from the systematic review by Zanganeh et al. [14]. “From their review” is sufficient.
- Table 4. Setting apart number and sources wuld enhance readability, , e.g.
2
[61,62]
rather than 2[61,62].
- “Studies that featured decision-analytic modelling tended to have a lifetime horizon”. Over longer horizons, the issues of adherence and attrition become important in the context of an intervention. Is there any information about them?
- “There is increasing evidence of the effectiveness of SSB tax policies as several jurisdictions around the world continue to implement population-wide policies aimed at improving food intake and reducing obesity. There is increasing evidence of the effectiveness of SSB tax policies.” First, this is repetitious. Second, “… as several jurisdictions continue …” could be read as a reason for effectiveness rather than an increase of available evidence.
- “ Recent research from the UK shows strong evidence that a soft drinks industry levy might limit exposure to liquid sugars and consequent health risks [137,138]. So there was no substitution of liquid sugar by other drinks or foods that may have (potentially even worse) effects on health? If substitution was tested for, a remark to this effect would be reassuring.
- The studies cited in the preceding para. seem to use a (0,1) indicator of obesity rather than a continuous variable indicating the extent of obesity such as the BMI. Is this impression correct?
- “on wages” Can be deleted.
- 273. “Obese blacks and females”. “They” is sufficient.
- “Lundborg and colleagues “They” is sufficient.
- “they further estimated “. Here, “the authors” would be better.
Section 4
- “the challenge with synthesising costs”. Write, “challenge of” or “challenge associated with”.
- “Two studies that examined the impact of childhood excess weight on healthcare utilisation found levels of endogeneity suggestive of underestimated health service use in previous studies that also investigated the impact of childhood excess weight [159,160]. “ This sentence is difficult to understand. Endogeneity here means that healthcare services influence excess weight, presumably reducing it. The relationship of interest,runs from excessive weight to the use of healthcare services. Being “too low”, the explanatory variable therefore contains a component that can be likened to a measurement error (as actually argued on line 340). Under standard assumptions, this leads to a weakening of the estimated relationship. So what is underestimated is the impact healthcare services have on excess weight – but why use itself should have been underestimated is not evident. Also, how was a “level of endogeneity” determined? Pertinent tests usually do not render a “level of endogeneity”.
- “For instance, decision-makers may be interested in estimating the impact on economic costs of hypothetical reductions in the prevalence of childhood obesity”. Yes, indeed; however, no “central estimate” of the strength of the relationship is offered, leaving policy-makers in the dark.
- “it may be worth considering if the tariff or population that valued the change in HRQoL is appropriate.” This statement is puzzling because a tariff typically is a levy imposed on foreign trade.
- Is there something to be learned by policy-makers? Why should they be interested in outcomes measured in utility terms?
- “However, we note a shift towards economic evaluations of treatment interventions in recent years, with only nine solely treatment interventions between 2001 and 2015 versus five between 2016 and 2020”. How about, “However, we note a shift towards treatment interventions in recent years, with but nine economic evaluations between 2001 and 2015 but five already between 2016 and 2020”.
- “Although the differences in approaches to economic evaluations across countries hinder comparability, Torbica et al. [168] argue that a ‘one size fits all’ approach is not feasible”. Why “Although”? Lack of comparability can be seen as (another) argument against “one size fits all”.
- “The decision regarding which childhood excess weight intervention to adopt fits within the purview of economic evaluation”. How about, “The objective of economic evaluation is to provide guidance for the choice of intervention targeting CEW”.
- “Ideally, the type of economic evaluation applied should reflect the analytical perspective, institutional context, and methodological guidance of the geographical jurisdiction.” What is “methodological guidance”? If anything is global, it is methodology. How about “Ideally, the type of economic evaluation applied should reflect a country´s view of the issue´s importance and institutional context.”
- “to levy nutritionally-poor foods”. Better to write, “impose a levy on …”
- “but is debatable if there is an age gradient with respect to the effect of childhood excess weight on cognitive performance.” Why exactly debatable? An age gradient per se means that the relationship between CEW and cognitive performance increases or decreases with age. It is silent about notably statistical significance.
- “This suggests the need for more studies with disaggregated analyses”. This is a rather weak policy implication. Could it be strengthened? For instance, which of the four strands has the most definite results? Which one should be especially fostered?
Appendices
- Readibility could be enhanced by inserting commas in Table A1, e.g. 218,633.
- Same for Tables A3 and A4.
Author Response
Reviewer 1
Reviewer general remarks: The principal contribution of this paper is to collect recent economic evidence on four topics: (1) economic costs associated with childhood excess weight (CEW); (2) associated health utilities with CEW; (3) economic evaluation of interventions targeting CEW; and (4) economic determinants and broader consequences of CEW. The authors are to be commended for their massive research effort. 2. An important weakness of the paper is the lack of clear conclusions. Readers tend to be lost in the details. E.g., it would be helpful to have a central estimate of the costs in strand (1), or even better, the most credible estimate according to the authors. 3. Also, a comparison with other public health issues (e.g., tobacco) would place the findings in perspective. 4. Another weakness is the paper´s style in several places (see detailed comments below).
Author response: We have revised the manuscript’s style and specified the pages and line numbers where the changes are made under the detailed comments section of this document. We have also emphasized the following concise points in the conclusion section of the manuscript (lines 902-910).
- Economic costs associated with CEW: The reviewed studies on economic costs establish a positive association between increasing CEW and economic costs. However, establishing a central estimate is hindered by considerable heterogeneity in the literature. Across studies, the mean annual excess direct costs associated with CEW ranged from £89 and £1,829, whilst the mean lifetime excess direct costs ranged from £1,494 to £13,260 (£ sterling, 2020 prices).
- Economic evaluations of interventions targeting CEW: In the context of health economic evaluation, preventive interventions have been the primary focus and the most likely to be demonstrated to be cost-effective, although recent years have seen a notable shift towards the economic evaluation of treatment approaches.
Detailed comments (line) Section 1
Reviewer comment (line 18): “childhood excess weight”. Maybe worthwhile to introduce the acronym CEW.
Author response: Thank you, we have introduced this acronym throughout the manuscript.
Reviewer comment (line 21): “Our ‘de novo’ searches identified six additional studies”. In what sense additional? Since some year?
Author response: Details of the time elapsed since the original reviews and our review are reported in the introduction and methods sections. We feel that specifying them in the abstract will limit our ability to include other salient points, given that the word count is only 200 words.
Reviewer comment (lines 45-47): “impose substantial health care and indirect costs on society”. At this point, it is crucial to distinguish costs that are borne by the children (and parents) themselves and external costs borne by society. From an economic point of view, only the latter may (not automatically do!) justify public intervention. A difficulty is costs that parent’s impose on their children. Using them for justifying public intervention within the family is a slippery slope however because parents burden their children with all sorts of cost (but benefits, too, which may suffer from external intervention). Some pertinent language would be valuable (without necessarily passing a judgment).
Author response: In response to the reviewer’s comment, we have added a sentence to clarify this.
Reviewer comment (lines 51-55): “The first, a systematic review by”. “The first, by” is sufficient. 54. “research strand 1)”. At this point, the four strands have not been identified in the body of the paper. Pertinent language needs to be moved up from line 93f.
Author response: In response to the reviewer’s comment, we have restructured the paragraphs accordingly.
Reviewer comment (line 63): “the study was constrained by only including studies that estimated costs over a lifetime horizon”. What is wrong with an estimate covering a person´s lifetime? Conversely, readers should be told the rate of discounting used in this study.
Author response: The review that is cited here only included studies that covered a lifetime horizon and therefore missed studies conducted over shorter time horizons. We believe that the inclusion of studies spanning shorter periods such as those that report annual excess costs over shorter time periods are also relevant when investigating the economic consequences of CEW. Lifetime costs are usually estimated by decision modelling, which requires making additional assumptions and introduces further uncertainties. Our review adds to this body of research by reporting a range of annual excess costs as well as lifetime excess costs associated with childhood excess weight (lines 655-665). Particularly, we note that the relationship between excess annual costs and lifetime costs is not linear. We report discount rates for individual studies in Table 2. The original review by Hamilton et al did not apply any discounting in deriving their estimates.
Reviewer comment (line 64): “stated by the Hamilton and colleagues”. Delete “the”.
Author response: We have deleted “the”.
Reviewer comment (line 76): “longitudinal study designs, which are crucial for exploring potential reverse causation.” For some readers at least, it would be helpful to offer more explanation. Also, the direction of causation may still not be determinable even through panel data as soon as the dependent variable (productivity) and the explanatory one (CEW) contain error terms that are correlated over time. An example would be parents´ low education.
Author response: We agree that longitudinal study designs may not necessarily explain the direction of causation and we now refer to this issue in the discussion section (lines 739-740).
Reviewer comment (line 82): “Most of the studies”. “Most of them” would avoid repetition.
Author response: We have replaced “studies” with “interventions”.
Reviewer comment (line 84): “The systematic review by Zanganeh et al. [14] excluded pharmacological and surgical interventions.” Again, “This review excluded pharmacological and surgical interventions” is sufficient.
Author response: We have revised this sentence in response to the reviewer’s comment.
Reviewer comment (line 93): “Segal et al. [9] omitted“. “The authors omitted” would be nicer.
Author response: We have revised this sentence in response to the reviewer’s comment.
Section 2
Reviewer comment (lines 108-109): “we provide our methods for updating”. Authors are unlikely to provide methods. Either write, “information about our methods” or “document our methods”.
Author response: We have revised this sentence in response to the reviewer’s comment.
Reviewer comment (line 114): “The previous review”. Write, “previously cited review”
Author response: We have revised this sentence in response to the reviewer’s comment.
Reviewer comment (line 117): “used by Hamilton et al. [12]”. Again, repetitive. Write, “by the authors”.
Author response: We have revised this sentence in response to the reviewer’s comment.
Reviewer comment (line 122): “the age upon study entry”. Preferable to write, “age at study entry”.
Author response: We have revised this sentence in response to the reviewer’s comment.
Reviewer comment (line 124): “in the English language.”. “in English” is sufficient.
Author response: We have revised this sentence in response to the reviewer’s comment.
Reviewer comment (line 180): by Zanganeh and colleagues [14]”.
Author response: We believe that by Zanganeh and colleagues [14] is not repetitive since this is previously cited in a separate paragraph.
Reviewer comment (line 181): “CHEERS statement”. Has this acronym been explained?
Author response: Yes, the acronym CHEERS is defined in lines 134-5.
Reviewer comment (line 212): “Evidence on the cost-effectiveness of an intervention”. “Evidence” is sufficient in this context.
Author response: We have revised this sentence in response to the reviewer’s comment.
Section 3
Reviewer comment (line 239): “of these papers”. “them” is sufficient – English has pronouns!
Author response: We have revised this sentence accordingly.
Reviewer comment (line 243): “or language of full text publication”. Presumably, “in the full text publication”.
Author response: We have revised this sentence in response to the reviewer’s comment.
Reviewer comment (line 254): “one year to a lifetime”. Write, one year to lifetime”
Author response: We have revised this sentence in response to the reviewer’s comment.
Reviewer comment (line 265): “The age ranges upon study entry captured children between”. This statement is not clear. Maybe, “Age at study entry varied between 2 and 18 years”?
Author response: We have revised this sentence accordingly.
Reviewer comment (line 275): “However, both approaches are umbrella terms. “However, these are umbrella terms” is sufficient.
Author response: We have revised this sentence accordingly.
Reviewer comment (line 282): Decision-analytic modelling”. “modelling” can be deleted.
Author response: We have changed this in response to the reviewer’s comment.
Reviewer comment (line 291): Three US-based studies [29,30,36] [38] reported various types”. This looks like four studies.
Author response: We have changed this to four studies in response to the reviewer’s comment.
Reviewer comment (Table 2): “Currency unit (Price year)”. Better two write, “Currency (at prices of the year x)”
Author response: In Table 2, we report the original unit of measure (currency, price date) from the individual studies. We have, however, introduced a footnote at the bottom of Table 2 clarifying this.
Reviewer comment (line 303): “Mean individual-level excess lifetime costs relative to healthy weight ranged from £1,494”. It is puzzling that nowhere is the cost per kg excess weight discussed. Should it not make a difference whether a person has 5 kg or 10 kg excess weight?
Author response: This cannot be assessed from the studies because the comparisons are based on weight classification (e.g., overweight versus healthy weight) rather than a continuous variable.
Reviewer comment (line 319): “Lightwood et al. [28] (2020 price year).” Write, “(at 2020 prices)”.
Author response: Where applicable, we now write “(£ sterling, 2020 prices)” throughout the document.
Reviewer comment (lines 320-328): “estimated” – six times in one para. -- a bit more variation would be nice.
Author response: We have updated the terminology in this paragraph in response to the reviewer’s comment.
Reviewer comment (line 333): “The age ranges upon study entry captured children between 3-18 years.” Unclear statement. Age does not “capture” anyone.
Author response: We have revised this sentence in response to the reviewer’s comment.
Reviewer comment (lines 339-345): “Sonntag et al. [32] estimated excess lifetime indirect costs due to excess weight per girl and boy at £2,586 and £4,451, respectively (2020 price year). While for boys direct and indirect costs estimated by the same authors in related studies were similar, for girls, the direct excess costs (£7,432) were nearly three times the indirect costs (2020 price year) in the German population.” Preferably this should follow directly after the first mention of the study on line 1. Also, are the low relative indirect costs for girls caused by their lower lifetime earnings?
Author response: We have restructured this paragraph accordingly. We have also briefly included a potential cause of lower indirect costs for girls in the discussion section (lines 344-345 and lines 669-672).
Reviewer comment (line 395): “The highest estimates of utilities for healthy weight”. This statement is ambiguous, it might relate to an extrapolation from actual to ideal weight. Better to write, “associated with healthy weight in the sample”.
Author response: We have revised this sentence in response to the reviewer’s comment.
Reviewer comment (lines 403-404): “They found that the association between BMI z-score and utilities was affected by age and was statistically significant in girls but the association was weak in boys at all ages”. Somewhat amazingly, the “raw differences” in utility are small. Adding a sentence or two stating that small differences may conceal substantial ceteris paribus effects of excess weight would be useful.
Author response: We believe that this is an interpretation of the evidence that does not belong in the results section. We do, however, return to this issue in the discussion (lines 734-735).
Reviewer comment (line 411): “32 studies from the systematic review by Zanganeh et al. [14]. “From their review” is sufficient.
Author response: We believe that reference to ‘Zanganeh et al.’ is more appropriate in this sentence.
Reviewer comment (Table 4): Setting apart number and sources would enhance readability, e.g.
2
[61,62]
rather than 2[61,62].
Author response: We have updated Table 4 accordingly.
Reviewer comment (lines 482-483): “Studies that featured decision-analytic modelling tended to have a lifetime horizon”. Over longer horizons, the issues of adherence and attrition become important in the context of an intervention. Is there any information about them?
Author response: We have added a sentence in the discussion section acknowledging the uncertainties surrounding assumptions about the persistence of treatment effects within decision-analytic studies (lines 808-809 and lines 846-848). We also report on this briefly in the results section (lines 504-505).
Reviewer comment (lines 545-549): “There is increasing evidence of the effectiveness of SSB tax policies as several jurisdictions around the world continue to implement population-wide policies aimed at improving food intake and reducing obesity. There is increasing evidence of the effectiveness of SSB tax policies.” First, this is repetitious. Second, “… as several jurisdictions continue …” could be read as a reason for effectiveness rather than an increase of available evidence.
Author response: We have restructured this sentence in response to the reviewer’s comment.
Reviewer comment (lines 550-552): “Recent research from the UK shows strong evidence that a soft drinks industry levy might limit exposure to liquid sugars and consequent health risks [137,138]. So, there was no substitution of liquid sugar by other drinks or foods that may have (potentially even worse) effects on health? If substitution was tested for, a remark to this effect would be reassuring.
Author response: We have now included a sentence on substitution behaviour in the discussion section (lines 881-882).
Reviewer question: The studies cited in the preceding para. seem to use a (0,1) indicator of obesity rather than a continuous variable indicating the extent of obesity such as the BMI. Is this impression correct?
Author response: Yes, the weight status categories that have been assessed are mostly discrete rather than continuous.
Reviewer comment (line 618): “on wages” Can be deleted.
Author response: We have deleted “on wages”.
Reviewer comment (line 624): “Obese blacks and females”. “They” is sufficient.
Author response: We have revised the sentence in response to the reviewer’s comment.
Reviewer comment (line 630): “Lundborg and colleagues “They” is sufficient.
Author response: We have revised the sentence in response to the reviewer’s comment.
Reviewer comment (line 636): “they further estimated “. Here, “the authors” would be better.
Author response: We have revised the sentence in response to the reviewer’s comment.
Section 4
Reviewer comment (line 685): “the challenge with synthesising costs”. Write, “challenge of” or “challenge associated with”.
Author response: We have revised this sentence in response to the reviewer’s comment.
Reviewer comment (lines 687-691): “Two studies that examined the impact of childhood excess weight on healthcare utilisation found levels of endogeneity suggestive of underestimated health service use in previous studies that also investigated the impact of childhood excess weight [159,160]. “This sentence is difficult to understand. Endogeneity here means that healthcare services influence excess weight, presumably reducing it. The relationship of interest runs from excessive weight to the use of healthcare services. Being “too low”, the explanatory variable therefore contains a component that can be likened to a measurement error (as actually argued on line 340). Under standard assumptions, this leads to a weakening of the estimated relationship. So, what is underestimated is the impact healthcare services have on excess weight – but why use itself should have been underestimated is not evident. Also, how was a “level of endogeneity” determined? Pertinent tests usually do not render a “level of endogeneity”.
Author response: In response to the reviewer’s comment, we have restructured this sentence to make it clearer.
Reviewer comment (lines 720-721): “For instance, decision-makers may be interested in estimating the impact on economic costs of hypothetical reductions in the prevalence of childhood obesity”. Yes, indeed; however, no “central estimate” of the strength of the relationship is offered, leaving policy-makers in the dark.
Author response: We highlight earlier in the discussion section the challenges associated with comparability of cost estimates and deriving a “central estimate” that is relevant to multiple jurisdictions (lines 673-685). There are pertinent methodological differences across the reviewed studies that would make such a central estimate questionable. In a systematic review of direct and indirect costs of excess weight by Dee et al. 2014, the authors came to similar conclusions.
(Dee, A., Kearns, K., O’Neill, C., Sharp, L., Staines, A., O’Dwyer, V., ... & Perry, I. J. (2014). The direct and indirect costs of both overweight and obesity: a systematic review. BMC research notes, 7(1), 1-9.).
Reviewer comment (line 763): “it may be worth considering if the tariff or population that valued the change in HRQoL is appropriate.” This statement is puzzling because a tariff typically is a levy imposed on foreign trade.
Author response: The word “tariff” is often used in similar contexts in health utility research, but we agree that it may be confusing for some readers and we have therefore deleted it.
Reviewer comment (lines 775-783): Is there something to be learned by policy-makers? Why should they be interested in outcomes measured in utility terms?
Author response: Decision-making bodies such as the National Institute of Health and Care Excellence (NICE) are often interested in the overall body of evidence surrounding health utility values since the uncertainty surrounding these outcomes can influence cost-effectiveness based decision-making.
Reviewer comment (lines 795-798): “However, we note a shift towards economic evaluations of treatment interventions in recent years, with only nine solely treatment interventions between 2001 and 2015 versus five between 2016 and 2020”. How about, “However, we note a shift towards treatment interventions in recent years, with but nine economic evaluations between 2001 and 2015 but five already between 2016 and 2020”.
Author response: We believe that the suggested change would not improve the clarity of the sentence.
Reviewer comment (lines 816-817): “Although the differences in approaches to economic evaluations across countries hinder comparability, Torbica et al. [168] argue that a ‘one size fits all’ approach is not feasible”. Why “Although”? Lack of comparability can be seen as (another) argument against “one size fits all”.
Author response: We have revised the sentence in response to the reviewer’s comment.
Reviewer comment (lines 851-853): “The decision regarding which childhood excess weight intervention to adopt fits within the purview of economic evaluation”. How about, “The objective of economic evaluation is to provide guidance for the choice of intervention targeting CEW”.
Author response: We have revised the sentence in response to the reviewer’s comment.
Reviewer comment (lines 853-855): “Ideally, the type of economic evaluation applied should reflect the analytical perspective, institutional context, and methodological guidance of the geographical jurisdiction.” What is “methodological guidance”? If anything is global, it is methodology. How about “Ideally, the type of economic evaluation applied should reflect a country´s view of the issue´s importance and institutional context.”
Author response: We have revised the sentence in response to the reviewer’s comment.
Reviewer comment (line 873): “to levy nutritionally-poor foods”. Better to write, “impose a levy on …”
Author response: We have revised the sentence in response to the reviewer’s comment.
Reviewer comment (line 885-886): “but is debatable if there is an age gradient with respect to the effect of childhood excess weight on cognitive performance.” Why exactly debatable? An age gradient per se means that the relationship between CEW and cognitive performance increases or decreases with age. It is silent about notably statistical significance.
Author response: The text referred to by the reviewer implies that the evidence from the different reviewed studies on an age gradient was limited and inconsistent.
Reviewer comment (line 898): “This suggests the need for more studies with disaggregated analyses”. This is a rather weak policy implication. Could it be strengthened? For instance, which of the four strands has the most definite results? Which one should be especially fostered?
Author response:
We have revised the manuscript to emphasize the following concise points in the conclusion section of the manuscript (lines 903-911).
- Economic costs associated with CEW: The reviewed studies on economic costs establish a positive association between increasing CEW and economic costs. However, establishing a central estimate is hindered by considerable heterogeneity in the literature. Across studies, the mean annual excess direct costs associated with CEW ranged from £89 and £1,829, whilst the mean lifetime excess direct costs ranged from £1,494 to £13,260 (£ sterling, 2020 prices).
- Economic evaluations of interventions targeting CEW: In the context of health economic evaluation, preventive interventions have been the primary focus and the most likely to be demonstrated to be cost-effective, although recent years have seen a notable shift towards the economic evaluation of treatment approaches.
Reviewer comment (appendices): Readability could be enhanced by inserting commas in Table A1, e.g., 218,633.
Author response: We have updated this table accordingly.
Reviewer comment (appendices): Same for Tables A3 and A4.
Author response: We have updated these tables accordingly.
Reviewer 2 Report
While I understand in general the purpose of these kinds of systematic reviews, in this case I am not clear about what the value-added is of this study. I read the author's justification and I am not convinced. In particular, in most cases, only very few papers are reviewed as a percentage of what is available because I believe the exclusion criteria are too strong. In another case, the author simply reiterates someone else's review and adds 2 more papers to their systematic review. This is also immensely LONG. The authors need to think more clearly about what they should and shouldn't include in the manuscript, sticking to summaries rather than a complete table for "every" paper. 171 pages is not a reasonable length to be published. For the contribution to the field, this paper should be "short", as I don't see that this is a particularly large contribution.
Author Response
Reviewer 2
Reviewer comment: While I understand in general the purpose of these kinds of systematic reviews, in this case I am not clear about what the value-added is of this study. I read the author's justification and I am not convinced. In particular, in most cases, only very few papers are reviewed as a percentage of what is available because I believe the exclusion criteria are too strong. In another case, the author simply reiterates someone else's review and adds 2 more papers to their systematic review. This is also immensely LONG. The authors need to think more clearly about what they should and shouldn't include in the manuscript, sticking to summaries rather than a complete table for "every" paper. 171 pages is not a reasonable length to be published. For the contribution to the field, this paper should be "short", as I don't see that this is a particularly large contribution.
Author response: We do not believe that our exclusion criteria are too strong. We worked with an information specialist to develop and pilot broad and inclusive search strategies. For two of the research strands reviewed, (1) economic costs associated with CEW and (2) economic evaluations of interventions targeting CEW, we broadened the inclusion criteria used in the earlier reviews. For instance, in (1) economic costs associated with CEW, our searches captured studies that reported excess annual costs which were excluded in the previous systematic review. In (2) economic evaluations of interventions targeting CEW, we expanded our inclusion criteria to include pharmacologic and bariatric surgery interventions.
Given the dearth of studies in the fourth research strand, namely the economic determinants and broader consequences of CEW, we believe that the additional studies identified by our searches, though few, are significant and relevant. For example, the previous systematic review identified only three studies on educational attainment and our search identified an additional study. For labour market outcomes, the previous systematic review identified only one study while our searches identified two articles consisting of four studies.
Regarding the length of this article, the manuscript consists of 27 pages, excluding the appendices and references. Perhaps it is more appropriate to refer to the appendices as supplementary materials and make it clear that though complementary, they will constitute a separate online resource that augments the main paper.
Reviewer 3 Report
This structural review contributes to the literature of childhood obesity studies. Please carefully consider the following three comments.
[1] Two studies that focused on the impact of childhood obesity on healthcare costs and used instrumental variables are shown. For the economic burden in Australia, Black et al. (2018) used a biological parent’s BMI as an instrument for the child’s BMI, and found that obese individuals incurred healthcare costs $102.90 AUD greater than their normal weight counterparts.
In section 4.1.1., the authors argued that the IV estimator is not without limitations.
Indeed, Biener et al. (2020) pointed out the possibility that the instrument is correlated with unobservables in the error term. They showed that mothers' access to and knowledge of the health care system does not drive the estimated effects for child BMI.
Please show the candidate variable as an additional IV variable. When researchers use the dataset that did not collect the biological parent’s BMI, the candidate variable may be useful.
[2] Please take account of the mendelian randomization (MR) and a genetic risk score. The MR study design reduces both reverse causation and confounding. Dick et al (2021) showed that no published studies had investigated the link between obesity and healthcare costs using MR methods.
[3] School-based interventions were cost-effective in boys or in girls are unclear. On the other hand, no previous studies have shown the underlying mechanisms of association between SES measures and childhood excess weight.
However, can we consider that "schools in disadvantaged areas" is associated with the cost-effectiveness of school-based interventions? Please add the discussion about it into the text.Black, N. et al. (2018) The health care costs of childhood obesity in Australia: An instrumental variables approach. Economics and Human Biology 31:1-13.
https://doi.org/10.1016/j.ehb.2018.07.003
Biener, AI. et al. (2020) The medical care costs of obesity and severe obesity in youth: An instrumental variables approach. Health Economics 29(5):1-16.
https://doi.org/10.1002/hec.4007
Dick, K. et al. (2021) Mendelian randomization: estimation of inpatient hospital costs attributable to obesity. Health Economics Review 11:16.
https://doi.org/10.1186/s13561-021-00314-2
Author Response
Reviewer 3
Reviewer comment: This structural review contributes to the literature of childhood obesity studies. Please carefully consider the following three comments.
Reviewer comment [1]: Two studies that focused on the impact of childhood obesity on healthcare costs and used instrumental variables are shown. For the economic burden in Australia, Black et al. (2018) used a biological parent’s BMI as an instrument for the child’s BMI and found that obese individuals incurred healthcare costs $102.90 AUD greater than their normal weight counterparts. In section 4.1.1., the authors argued that the IV estimator is not without limitations. Indeed, Biener et al. (2020) pointed out the possibility that the instrument is correlated with unobservables in the error term. They showed that mothers' access to and knowledge of the health care system does not drive the estimated effects for child BMI.
Please show the candidate variable as an additional IV variable. When researchers use the dataset that did not collect the biological parent’s BMI, the candidate variable may be useful.
Author response: The only other candidate variable explored as an additional IV variable by Biener et al. (2020) was stepparents' BMI as an instrument for child weight in a falsification test. They found that this was not a powerful instrument and we therefore do not refer to it in the paper.
Reviewer comment [2]: Please take account of the mendelian randomization (MR) and a genetic risk score. The MR study design reduces both reverse causation and confounding. Dick et al (2021) showed that no published studies had investigated the link between obesity and healthcare costs using MR methods.
Author response: The study by Dick et al (2021) is based on an adult population of participants between the ages of 40 and 69 using inherited genetic variants as the instrument. However, we have now included a sentence referencing it and recommending it for future studies in childhood populations (line 715-716).
Reviewer comment [3]: School-based interventions were cost-effective in boys or in girls are unclear. On the other hand, no previous studies have shown the underlying mechanisms of association between SES measures and childhood excess weight.
However, can we consider that "schools in disadvantaged areas" is associated with the cost-effectiveness of school-based interventions? Please add the discussion about it into the text.
Author response: It is difficult to draw a meaningful conclusion about the association between school-based interventions in disadvantaged areas and cost-effectiveness because very few economic evaluations in this strand were stratified by socio-economic status. However, we have recommended subgroup analysis as a way of assessing possible variation in cost-effectiveness estimates by predictive factors (lines 844-846).
Round 2
Reviewer 1 Report
Line 691. "A major cause of endogeneity (known as omitted variable bias ..."
This is not correct. Endogeneity of an explanatory variable x means that it depends on the variable y to be explained. Therefore, x contains a stochastic error component that is correlated with the error component u of y, necessarily violating the requirement that x and u are uncorrelated of regression analysis.
Omitted variable bias is something different. In this case, the error component u of y is augmented by the omitted variable x´. To the extent that x´ is correlated with one of the included explanatory variables x (whith is not ncessarily the case), this induces a correlation between x and the augmented error term, violating the requirement of no correlation between x and u of regression anlysis.
Author Response
Reviewer 1 Round 2
Reviewer comment (Line 649): "A major cause of endogeneity (known as omitted variable bias ..."
This is not correct. Endogeneity of an explanatory variable x means that it depends on the variable y to be explained. Therefore, x contains a stochastic error component that is correlated with the error component u of y, necessarily violating the requirement that x and u are uncorrelated of regression analysis.
Omitted variable bias is something different. In this case, the error component u of y is augmented by the omitted variable x´. To the extent that x´ is correlated with one of the included explanatory variables x (which is not necessarily the case), this induces a correlation between x and the augmented error term, violating the requirement of no correlation between x and u of regression analysis.
Author response: We have revised the paragraph containing the sentence in response to the reviewer’s comment (lines 647-651).
Reviewer 2 Report
Nothing was changed in reference to my comments, and I don't believe this is a value addition to the literature. My statement remains REJECT.
Author Response
No response required.